# Visualizing endogenous opioid receptors in living neurons using ligand-directed chemistry

Seksiri Arttamangkul[1]*, Andrew Plazek[2], Emily J Platt[3], Haihong Jin[2], Thomas F Murray[4], William T Birdsong[1†], Kenner C Rice[5], David L Farrens[3], John T Williams[1]

[1]The Vollum Institute, Oregon Health & Science University, Portland, United States; [2]Medicinal Chemistry Core, Oregon Health & Science University, Portland, United States; [3]Department of Biochemistry and Molecular Biology, School of Medicine, Oregon Health & Science University, Portland, United States; [4]Department of Pharmacology, School of Medicine, Creighton University, Omaha, United States; [5]Drug Design and Synthesis Section, Intramural Research Program, NIDA and NIAAA, Bethesda, United States

**Abstract** Identifying neurons that have functional opioid receptors is fundamental for the understanding of the cellular, synaptic and systems actions of opioids. Current techniques are limited to post hoc analyses of fixed tissues. Here we developed a fluorescent probe, naltrexamine-acylimidazole (NAI), to label opioid receptors based on a chemical approach termed 'traceless affinity labeling'. In this approach, a high affinity antagonist naltrexamine is used as the guide molecule for a transferring reaction of acylimidazole at the receptor. This reaction generates a fluorescent dye covalently linked to the receptor while naltrexamine is liberated and leaves the binding site. The labeling induced by this reagent allowed visualization of opioid-sensitive neurons in rat and mouse brains without loss of function of the fluorescently labeled receptors. The ability to locate endogenous receptors in living tissues will aid considerably in establishing the distribution and physiological role of opioid receptors in the CNS of wild type animals.
DOI: https://doi.org/10.7554/eLife.49319.001

*For correspondence:
arttaman@ohsu.edu

Present address: †Department of Pharmacology, University of Michigan, Ann Arbor, United States

**Competing interests:** The authors declare that no competing interests exist.

## Introduction

Since the discovery of receptors for opiate drugs in early 1970 s, studies have focused on opioid actions in normal physiological and pathological conditions. Through pharmacological and cloning experiments, three opioid receptor subtypes: μ (MOR), δ (DOR) and κ (KOR) have been established (*Kieffer and Evans, 2009*). These receptors are found throughout the central and peripheral nervous systems with unique distributions for each subtype. To date, the methods used for identifying these receptors in brain slices have relied on immunohistochemistry or genetically modified receptors. Fluorescent ligands based on opioid peptides and opiate alkaloids have been developed and used to visualize the receptors of cultured cells. Unfortunately, the use of these small-molecule ligands is often limited by the decrease in affinity and selectivity that results from the addition of a fluorescent molecule that is equivalent to or larger in size than the opioid ligand (*Archer et al., 1992*; *Gaudriault et al., 1997*; *Madsen et al., 2000*; *Arttamangkul et al., 2000*; *Lam et al., 2018*). More-over, these fluorescent ligands possess pharmacological actions and bind reversibly to the receptors, thus making them less useful for labeling receptors that later on will be studied for biological activity. Accordingly, a new strategy for long-lasting labeling of receptors without altering naïve receptor functions is desirable.

Affinity labeling has been a very useful technique to chemically link target-molecules to specific proteins (*Takemori and Portoghese, 1985*). During the past decade, Hamachi and colleagues have developed a chemical method termed, 'traceless affinity labeling' to covalently tag a fluorescent dye to the protein of interest with the use of ligand guidance (*Figure 1A*; *Hayashi and Hamachi, 2012*). The key advantage of this approach is that once the labeling reaction occurs, the guide ligand is free to dissociate while leaving the fluorescent probe covalently attached to the protein. In practice, a high affinity ligand is used to direct probe binding and ensure the labeling occurs near the ligand binding site. The close proximity between the ligand binding pocket and the reactive sites effectively increases the concentration of the reactive groups, enabling a probe of mild reactivity to be used. Because of the high affinity ligand guide, specific labeling can be obtained with low concentrations thus minimizing non-specific background reactions. Several of these ligand-directed molecules have been studied (*Tamura and Hamachi, 2019*). Among them, the reactive probe of electrophilic acylimidazole (AI) has been shown to label a number of membrane-bound proteins, including folate, AMPA and GABA$_A$ receptors in living cells (*Fujishima et al., 2012*; *Yamaura et al., 2016*; *Wakayama et al., 2017*).

In the present study, a fluorescent ligand for opioid receptors based on the ligand-directed acylimidazole approach was developed (*Figure 1*). β-naltrexamine was chosen as the guide molecule (pharmacophore of the binding site) because it (1) has high binding affinity for all opioid receptor subtypes, (2) has been used for conjugation with fluorescent dyes with minimal reduction in affinity (*Emmerson et al., 1997*) and (3) is the parent compound of β-funaltrexamine (β-FNA), an affinity-labeling analog that irreversibly binds to MOR (*Portoghese et al., 1980*). Fluorescent dyes were then linked to the naltrexamine-acylimidazole (NAI) parent compound via click chemistry. This small molecule was able to penetrate and label receptors deep within living brain slices. The labeling was stable following wash or application of naloxone. Fluorescently labeled endogenous receptors were observed in the locus coeruleus and several other areas in living brain slices from both rats and mice. Importantly, following the labeling process, the labeled neurons remained responsive to agonists.

## Results

### Synthesis of fluorescent naltrexamine-acylimidazole

The fluorescent naltrexamine-acylimidazole was designed based upon the crystal structure of β-funaltrexamine (β-FNA) bound MOR (*Manglik et al., 2012*). The key covalent attachment of β-FNA at K233 side-chain was purposely avoided. The linker for NAI was designed to be longer than that for β-FNA and was predicted to chemically modify nucleophilic side chains of amino acids on the extracellular loops 2 and 3 (*Figure 1—figure supplement 1*). The compound was made in two steps. First, naltrexamine-acylimidazole alkyne (NAI-AK) was synthesized. The β-epimer of naltrexamine was stereospecifically derived from a non-selective opioid antagonist naltrexone (*Sayre and Portoghese, 1980*). Acylimidazole was incorporated into the naltrexamine to produce a proper linker and a terminal alkyne (*Figure 1B*). The second part of the synthesis was designed for coupling an azide probe to NAI-AK via copper-catalyzed azide-alkyne cycloaddition known as click chemistry (*Figure 1C*; *Tornøe et al., 2002*; *Rostovtsev et al., 2002*). The click reaction was optimized to be in slightly acidic conditions (pH ≈ 4) to prevent hydrolysis of the acylimidazole linker. Incorporation of Alexa 594 azide to NAI-AK gave NAI-A594 (*Figure 1C*) with a good yield and high purity following high performance liquid chromatography. Competitive radioligand binding assays showed that NAI-A594 bound to MOR, DOR and KOR with Ki ≈ 50, 70 and 200 nM, respectively (n = 2 of triplicate samples, see *Table 1*). NAI-A488 was prepared as described for NAI-A594.

### NAI-A594 covalently labeled MORs in HEK293 cells

HEK293 cells stably expressing FLAG-epitope tagged MORs (FMOR) were first used to test the ability of NAI-A594 to transfer the fluorophore to the receptor. In the first experiment, the stability of NAI-A594 labeling on living cells was visualized with a spinning disc confocal microscope. FMOR-expressing cells were labeled with NAI-A594 and as a control, co-labeled with anti-FLAG (M1) antibody conjugated to Alexa 488 (M1-A488). Confocal images revealed the fluorescence of A594 and M1-A488 along the plasma membrane even after washing with the high affinity antagonist, naloxone (*Figure 2A–a*). For comparison, the reversible fluorescent ligand, naltrexamine-Alexa594 (NTX-A594,

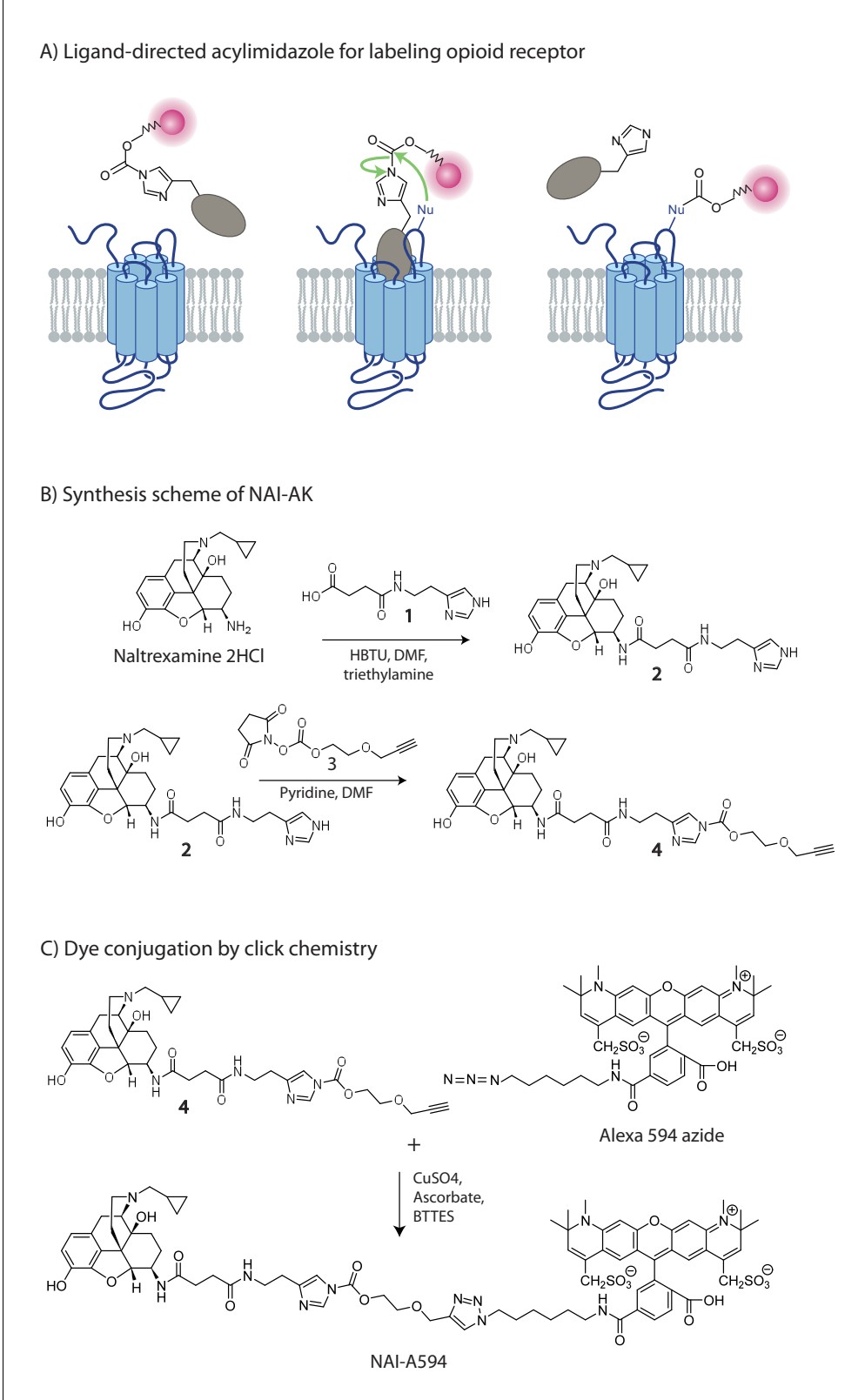

**Figure 1.** The chemistry of fluorescent naltrexamine-acylimidazole (NAI). (**A**) A schematic showing the steps in fluorescent dye attachment to the opioid receptor via ligand-directed acylimidazole chemistry. The ligand (gray) represents naltrexamine and an appropriate length of linker. The red circle represents a fluorescent dye. Left panel shows naltrexamine guides the molecule to the receptor, and interacts (middle panel) with the binding pocket.

*Figure 1 continued on next page*

*Figure 1 continued*

The carbonyl of acylimidazole can therefore react with a nucleophilic side-chain of an amino acid that is Lys, Ser, Tyr and Thr (*Fujishima et al., 2012*). Once the transfer reaction occurs the dye remains attached to the receptor while naltrexamine moiety is free to dissociate. (right panel) (**B**) The synthesis scheme of the naltrexamine-acylimidazole alkyne (NAI-AK) precursor. The linker arm of alkyne is introduced for conjugation with dye via click chemistry. (**C**) Illustration of the click reaction to yield NAI-A594.

DOI: https://doi.org/10.7554/eLife.49319.002

The following figure supplement is available for figure 1:

**Figure supplement 1.** Hypothetical sites for NAI modification.

DOI: https://doi.org/10.7554/eLife.49319.003

$K_d$ ~53 nM, *Birdsong et al., 2013*) was used for labeling instead of the NAI-A594. The fluorescence of A594 but not of M1-A488 on plasma membrane was almost completely lost after the naloxone-wash (*Figure 2A–b*).

Given that NAI-A594 also binds to DOR and KOR in nanomolar concentrations, experiments using FLAG-DOR (FDOR, stably expressed) and FLAG-KOR (FKOR, transiently expressed) in HEK293 cells were done (*Figure 2—figure supplement 1*). After washing with naloxone (1 µM) the fluorescence of NAI-A594 on FDOR expressing cells was comparable to that in FMOR expressing cells. Persistent labeling of FKOR was negligible.

Next, flow cytometry was used to confirm in FMOR cells that NAI-A594 efficiently labels the receptors. The average fluorescence intensity of cells labeled with NAI-A594 (100 nM, 1 hr at 37°C) was similar to antibody labeling using M1-A594 despite the fact that antibodies had an average four dye molecules per protein. Application of naloxone (10 µM) decreased the average fluorescence intensity by 30%, suggesting that there was a reversible component of NAI-594 labeling (*Figure 2B*). When naloxone was co-incubated during the labeling of NAI-A594 (100 nM), the average fluorescence intensity was reduced to only 2% of the NAI-A594 treatment alone, indicating that non-specific staining was negligible.

Under the same conditions the fluorescence intensity of cells labeled with the non-covalently labeling probe NTX-A594 (100 nM) was examined. The fluorescence intensity was only 40% of that with NAI-A594 and was further reduced with the application of naloxone. As the concentration of NAI-A594 increased, the fluorescence labeling also increased (*Figure 2B*).

Biochemical experiments were also carried out to verify that the dye moiety was in fact irreversibly bound to the receptors. In one set of experiments, NAI-A488 was synthesized, and used to label cells for subsequent immunoprecipitation and Western blot analysis (*Figure 2C*). In brief, FMOR cells were incubated with NAI-A488 (100 nM) in the absence (lane 4) and presence of naloxone (lane 3). Unlabeled FMOR cells (lane 2) and HEK293 cells not expressing FMOR (lane 1) were also included in the assays. Next, cell extracts were prepared and FMOR was immunoprecipitated using a rabbit anti-MOR antibody (UMB3) followed by western blotting with either rabbit anti-A488 (*Figure 2C*, left gel) or anti-FLAG mouse monoclonal (M1, *Figure 2C*, right gel). Immunoblotting with anti-A488

**Table 1.** Competitive binding assays in CHO cells expressing MOR, DOR or KOR receptors.
Data were reported as $K_i$ in nM from two independent experiments performed in triplicate.

| | MOR [³H]DAMGO | DOR [³H]DPDPE | KOR [³H]diprenorphine |
|---|---|---|---|
| Naltrexamine | 0.26 ± 0.02[a] | 117.06 ± 8.94[a] | 5.15 ± 0.26[a] |
| NAI-A594 | 60.36, 53.03 | 81.03, 64.93 | 210.8, 208.2 |
| DAMGO | 0.393, 0.691 | | |
| DPDPE | | 0.747, 0.685 | |
| DynA(1-13)NH₂ | | | 0.973, 0.756 |

DAMGO, DPDPE and DynA(1-13)NH₂ were tested as positive controls under the same conditions. Each data point was done in triplicate samples. [a] Reference: *Li et al. (2009)*.

DOI: https://doi.org/10.7554/eLife.49319.004

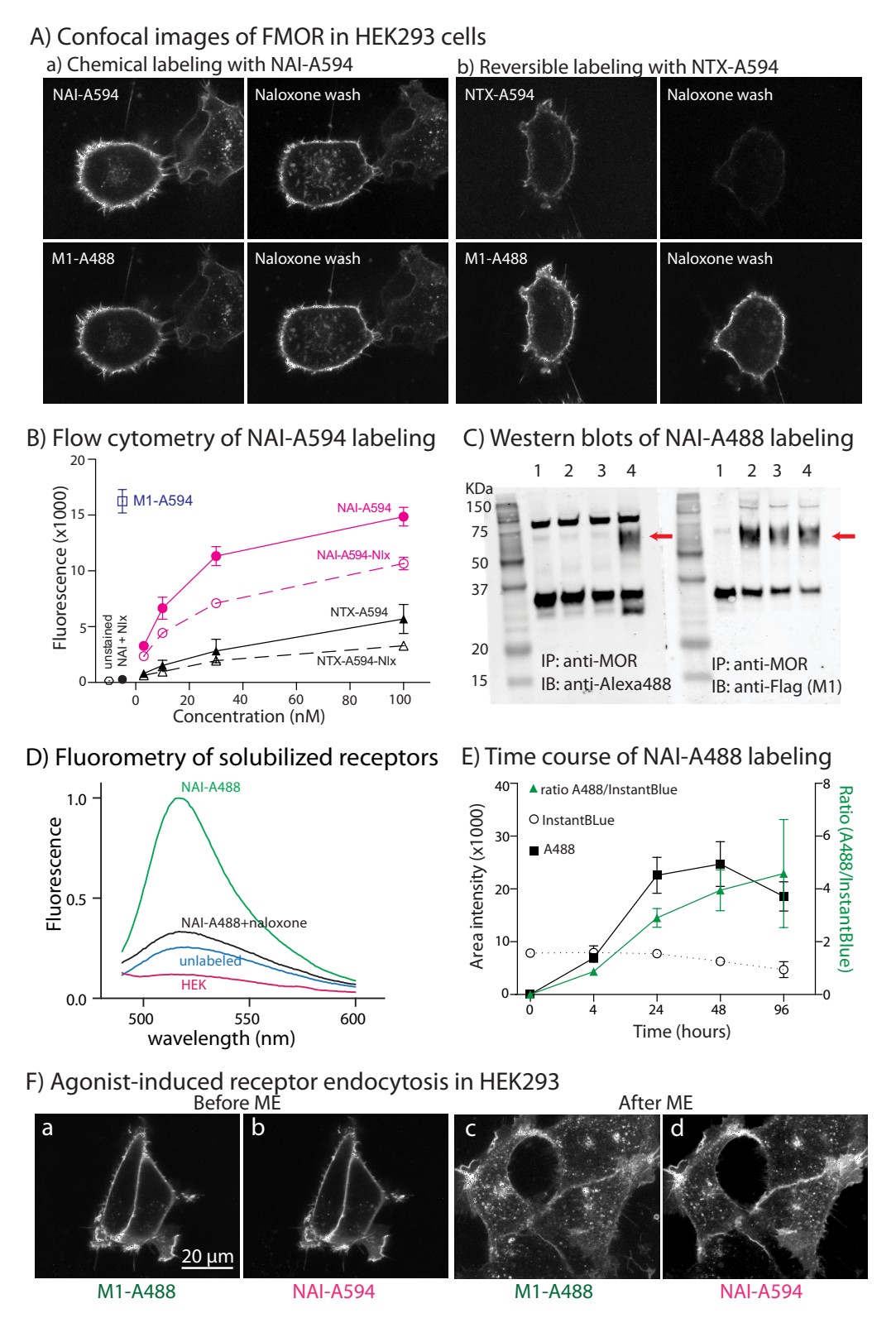

**Figure 2.** Naltrexamine-acylimidazole modified with fluorescent dyes (NAI-A594 or NAI-A488) labels FMORs in HEK293 cells. (**A**) Micrographs showing fluorescent labeling of cell using NAI-A594 and M1-A488. Images in (**a**) demonstrate that following labeling of receptors with NAI-A594, the treatment with naloxone (1 μM, 10 min, n = 3) did not reduce the fluorescence along the plasma membrane. Images in (**b**) show the fluorescence of A594 from the reversible ligand NTX-A594 labeling was weaker and abolished following treatment with naloxone (1 μM, 10 min, n = 3). In both (**a**) and

*Figure 2 continued on next page*

*Figure 2 continued*

(b), the fluorescence of antibody labeling using M1-A488 wasere not changed after the treatment of naloxone. (B) Flow cytometry shows an increase in fluorescence with increasing concentrations of NAI-A594 and NTX-A594. The red line shows the signal after wash with a buffer [2 × 2 mL DMEM (no phenol red) plus 5% FBS, room temperature]. The red dashed-line shows fluorescent signal after wash with naloxone (10 µM, 2 × 2 mL, room temperature, n = 3). The black lines (both solid and dashed, n = 3) represent cells treated with NTX-A594 and washed as described for NAI-A594 treated cells. When cells were co-incubated with naloxone (10 µM), there was negligible NTX-A594 fluorescence (black dot). The fluorescence of antibody labeling M1-A594 was also quantified as shown with a blue square. (C) Immunoprecipitation-western blot analyses show the labeling of FMOR with NAI-A488 (100 nM, 1 hr, 37°C, 5% CO2). The solubilized receptors from these cells were immunoprecipitated with rabbit anti-MOR and then immunoblotted with primary rabbit anti-Alexa488 and secondary goat-anti rabbit Alexa-680 (lanes 1–4, left panels) or primary M1 anti-FLAG and secondary goat-anti mouse Alexa-680 (lanes 1–4, right panels). lane 1: HEK293 cells that lack FMOR, lane 2: unlabeled FMOR cells, lane 3: FMOR cells labeled with NAI-A488 in the presence of 10 µM naloxone, and lane 4: NAI-A488 labeled FMOR cells. The experiments were repeated twice. (D) The fluorescent spectra of solubilized receptors in (C) were measured between wavelengths of 450–600 nm. Only the receptors that were incubated with NAI-A488 alone showed significant fluorescence. The fluorescence was blocked by 10 µM naloxone, was at a low level from unlabeled FMOR cells and was not observed in HEK cells without FMOR. (E) The labeling of purified FMOR by NAI-A488 increased with time. FMOR was purified and incubated in NAI-A488 at room temperature for the indicated time. Band intensities were measured using image J. The error bars represented standard deviation of two independent experiments. (F) Micrographs of cells co-labeled with NAI-A594 and M1-A488. Images show the plasma membrane staining of M1-A488 (a) and NAI-A594 (b) and the appearances of fluorescent puncta in M1-A488 (c) and NAI-A594 (d) labeled cells after application of 10 µM ME for 10 min.

DOI: https://doi.org/10.7554/eLife.49319.005

The following figure supplements are available for figure 2:

**Figure supplement 1.** NAI-A594 labels FMORs, FDORs and FKORs in HEK293 cells.

DOI: https://doi.org/10.7554/eLife.49319.006

**Figure supplement 2.** Labeling of MOR with NAI-594 and NAI-488.

DOI: https://doi.org/10.7554/eLife.49319.007

---

detected a ~ 75 kDa band of FMOR only in cells labeled with NAI-A488 in the absence of naloxone (*Figure 2C*, left gel, lane 4), indicating that A488 was irreversibly bound to FMOR. In contrast, the anti-FLAG antibody detected FMOR in all treatment conditions except the control HEK293 cells, verifying that FMOR was present in unlabeled as well as labeled cells. (*Figure 2C*, right gel, lanes 2–4). Nonspecific bands near 35 and 100 kDa were observed in every sample including the control HEK293 cells (*Figure 2C*, lane 1–4 of left and right panels). The fluorescence spectra of the cell extracts used in this experiment confirmed the results from the immunoprecipitation and Western blot analysis and displayed maximal emission at 517 nm, characteristic of A488 dye (*Figure 2D*). Fluorometry also confirmed that labeling of cells with NAI-A594 was FMOR-specific, and the emission maximum of 615 nm was consistent with the A594 fluorophore (*Figure 2—figure supplement 2A*).

The NAI compound was also used to directly label purified receptors. In this experiment, the purified FMOR (0.75 µM) was incubated with NAI-A488 (4.7 µM) at room temperature in the presence or absence of naloxone (100 µM). Aliquots were removed at 0, 4, 24, 48, and 96 hr and denatured by addition of SDS-sample buffer. Analysis of in-gel fluorescence showed that the attachment of A488 to receptors increased with time and reached maximal intensity after 48 hr (*Figure 2E*). No significant fluorescence was found in preparations co-incubated with naloxone, indicating that this labeling was specific to FMOR (*Figure 2—figure supplement 2B*).

Agonist-induced internalization of receptor could also be followed in cells that were treated with NAI-A594. FMORs labeled with NAI-A594 were found co-localized with M1-A488 in the cytoplasm after [Met⁵]enkephalin (ME) application, indicating that the A594 labeled receptors could bind agonist, internalize, and be visualized in the endosomes (*Figure 2F,a–d*).

## Labeling endogenous receptors in brain slices

Several brain areas were selected to evaluate NAI-A594 labeling of receptors on living neurons from rats and mice, as discussed below.

### Locus Coeruleus

Locus coeruleus (LC) neurons were first selected to test the labeling efficiency and functionality of the receptors after labeling. These noradrenergic neurons are an excellent model because they express only the MOR subtype and are a near homogenous population of densely packed neurons.

Brain slices containing LC neurons were prepared and incubated in a solution of NAI-A594 (100 nM) in oxygenated artificial cerebrospinal fluid (ACSF) at 30°C for 1 hr. Following incubation, macroscopic images showed fluorescent signals in the area of LC (*Figure 3A–a*). At higher magnification, single LC neurons were identified (*Figure 3A–a'*). With the use of a 2-photon microscope, fluorescence of Alexa594 was observed along the plasma membrane of LC neurons and on many processes (*Figure 3B–a*). Fluorescence remained after application of naloxone (10 µM, 10 min) indicating that the labeling was stable (*Figure 3B–a'*). Quantification of cell-associated fluorescence revealed a 40% decrease in the intensity of labeling after application of naloxone (*Figure 3—figure supplement 1A*). In contrast, labeling with the reversible ligand, NTX-A594, was not efficient and the fluorescence along the plasma membrane was nearly abolished after washing with ACSF for 10 min (*Figure 3B–b,b'*). There was no staining of NAI-A594 in LC neurons from MOR knockout rats (*Figure 3B–c,c'*). To further confirm the specific labeling of NAI-A594, slices from mice expressing green fluorescence protein under regulation of tyrosine hydroxylase promotor (THGFP) were used as a marker for the noradrenergic LC neurons. The staining of NAI-A594 was observed on the plasma membrane of both somata and dendrites of GFP expressing LC cells (*Figure 3B–d*). The labeling of NAI-A594 in the GFP positive neurons was blocked when slices were co-incubated with CTAP (1 µM), a MOR-selective antagonist (*Figure 3B–d'*). The fluorescence puncta found in these slices are autofluorescence from natural substrates presented in the neurons. These puncta were observed in both GFP and non-GFP neurons (*Figure 3—figure supplement 2*).

In the same horizontal slices that contained the LC neurons, strong fluorescent labeling with NAI-A594 was also found in the nearby lateral parabrachial nucleus (PB, *Figure 3A–a*). The labeling of a single neurons in this area was visualized with the 2-photon microscope (*Figure 3C–a*). Neurons in PB area labeled with NAI-A488 were also observed using the spinning disc confocal microscope (*Figure 3C–b*). Lateral to the pons, labeled neurons in hippocampal formation (HC) were also clearly visualized (*Figure 3A–a*). The fluorescent pattern in the hippocampus was very similar to that observed using MOR-specific immunohistochemistry in fixed tissue (*Arvidsson et al., 1995a*).

Labeling of endogenous MORs with NAI-A594 was both concentration- and time-dependent (*Figure 3—figure supplement 1B and C*). After incubation for 1 hr, labeling was observed with NAI-A594 at concentrations as low as 10 nM. Increasing the concentration to 100 nM resulted in a well-defined florescent signal along the plasma membrane. The time course of labeling was examined using NAI-A594 at a concentration of 30 nM. Incubation times from 0.5 to 2 hr showed an increase in staining. Brian slices that were left in the incubation vials longer than 2 hr did not result in additional increased fluorescence (data not shown). This rapid labeling at low concentrations makes NAI-A594 suitable for identifying receptors in living slices that will be used for functional experiments.

A distinct advantage for using the small molecule labeling approach is that the NAI compounds can penetrate and label receptors on the neurons deep into the slice, thus enabling detection of healthy neurons in three-dimensions. The penetration and labeling of the NAI-A594 was examined using brain slices from a transgenic animal expressing FMORs in LC. The same neurons were labeled with M1-A488 and NAI-A594 for comparison. LC neurons having FMORs labeled with NAI-A594 were observed deep (90 µm) in the slice at a depth where visualization of FMORs labeled with M1-A488 was not well-defined (*Figure 3D–a,b*, *Figure 3—figure supplement 1D,a* for quantification, and *Video 1* and *Video 2*). M1-594 labeling was also investigated in LC slices and compared to separate LC slices labeled with NAI-A594. The depth of labeling using M1-A594 was similar to that of M1-A488, less than the labeling of NAI-A594. (*Figure 3—figure supplement 1D,b* for quantification, and *Video 3* and *Video 4*).

## Labeling of NAI-A594 in other brain areas

Other brain areas comprising heterogeneous populations of neurons and expressing different opioid receptor subtypes were examined to evaluate the potential labeling with NAI-A594. Thus far, the results indicate that NAI-A594 was capable of stably labeling FMORs and FDORs in HEK293 and endogenous MORs in LC neurons. The next step was to investigate whether NAI-A594 chemically reacted with the endogenous DOR and KOR in brain slices. The labeling of NAI-A594 in several areas of the brain was thus studied and compared with the previously reported distribution of DORs and KORs (*Mansour et al., 1987*; *Tempel and Zukin, 1987*; *Arvidsson et al., 1995a*;

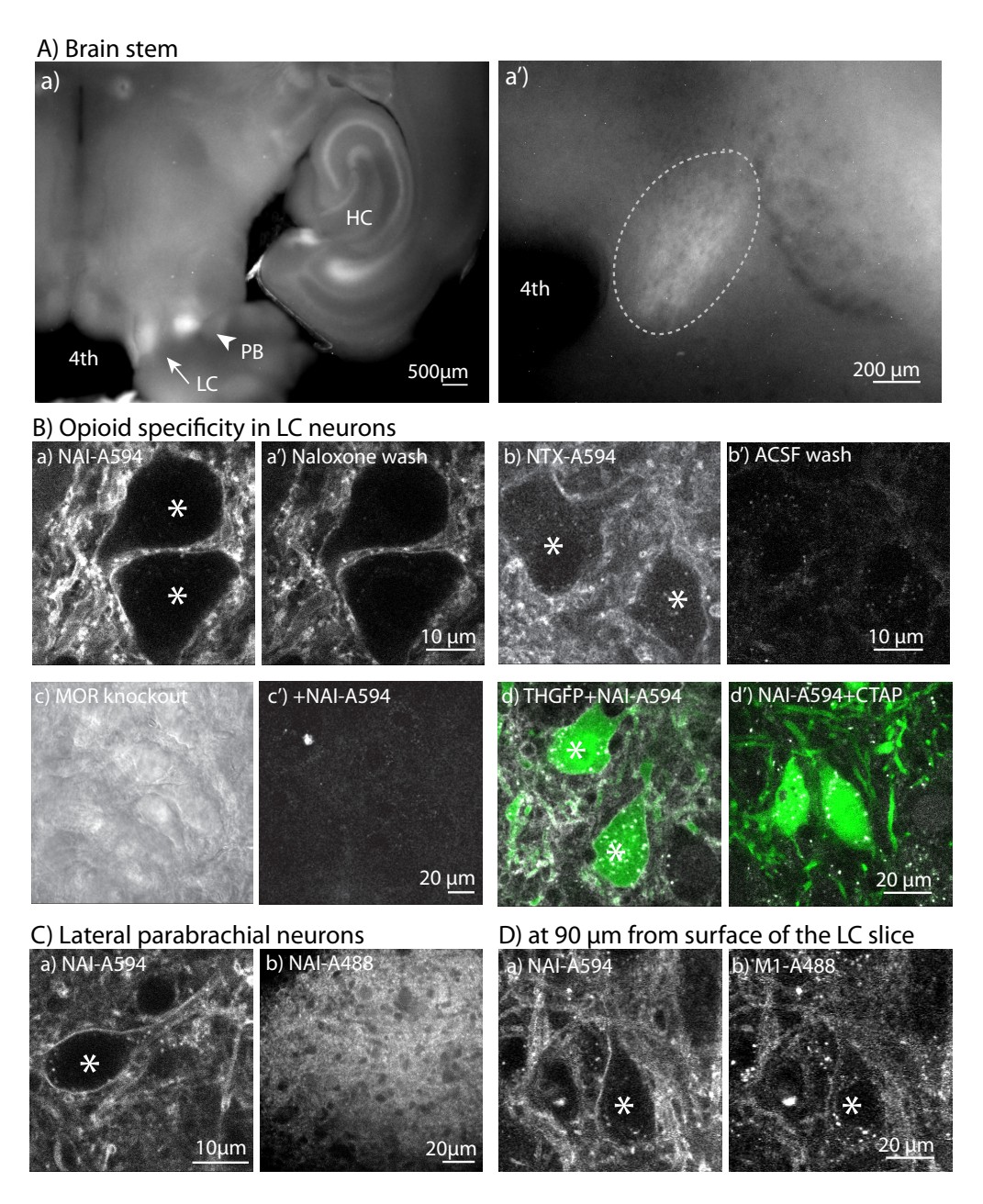

**Figure 3.** NAI-A594 labeled endogenous MORs on LC neurons. (**A**) Top panels show a low-magnification fluorescence image after incubating LC slices from wild type rats in NAI-A594 (100 nM in ACSF, 1 hr, 30℃). (**a**) the LC, parabrachial nucleus (PB) and parts of the hippocampus (HC) are labeled. a', a zoom-in image of the LC. (**a'**) the zoom-in picture shows A594 fluorescence of LC neurons (elliptical dash-line). (**B**) The 2-photon images of LC neurons stained with NAI-A594 or NTX-A594. The plasma membrane staining of LC neurons labeled with NAI-A594 was observed (**a**). The fluorescent signal on these neurons was reduced but remained on the cells after superfusion of naloxone (10 μM, 10 min, **a'**). When neurons were labeled with NTX-A594 (100 nM), there was a dim fluorescence on the plasma membrane (**b**) and the fluorescent signal was eliminated by a 10 min superfusion of ACSF (**b'**). Labeling of LC neurons with NAI-A594 was not observed in slices from a MOR knockout rat (**c**, bright field view and **c'**, fluorescence following incubation with NAI-A594). The labeling of NAI-A594 was blocked with MOR antagonist CTAP as shown by slices containing the LC from TH-GFP mice that were incubated with NAI-A-594 alone (**d**) and in combination with CTAP (1 μM, **d'**). (**C**) Images of the parabrachial nucleus incubated with NAI-A594 using a 2-photon microscope (**a**) and the PB neurons labeled with NAI-A88 using a spinning disc confocal microscope (**b**). (**D**) Images of the same slice containing the LC co-incubated in NAI-A594 (100 nM) and M1-A488 (10 μg/ml, 1 hr at 30℃). (**a**) shows the NAI-A594 labeling of neurons and processes at 90 μm below the surface. (**b**) shows the same z-optical section for M1-A488 labeling. *=Labeled neurons.

DOI: https://doi.org/10.7554/eLife.49319.008

*Figure 3 continued on next page*

*Figure 3 continued*

The following figure supplements are available for figure 3:

**Figure supplement 1.** Fluorescence in brain slices.

DOI: https://doi.org/10.7554/eLife.49319.009

**Figure supplement 2.** NAI-A594 labeled LC neurons from THGFP mice.

DOI: https://doi.org/10.7554/eLife.49319.010

*Arvidsson et al., 1995b*; *Ding et al., 1996*). Coronal and horizontal brain slices were prepared from rats or mice and the distribution of NAI-A594 labeling was examined.

## Striatum and cerebral cortex

All three opioid receptor subtypes are found within the striatum and nucleus accumbens (*Arvidsson et al., 1995b*; *Miura et al., 2007*; *Banghart et al., 2015*). MOR is present predominantly in the striosomes or patches. In contrast, DOR and KOR are diffusely distributed throughout the region. In coronal slices of striatum from mice, NAI-A594 labeling revealed a distinct distribution of fluorescent patches (*Figure 4-a,d*). Labeling was blocked when slices were co-incubated with CTAP (1 μM, *Figure 4-b*) indicating that NAI-A594 primarily labeled MORs. This distinctive patch distribution was also observed in coronal and horizontal slices of wild type rats (*Figure 4-c,e*). Labeling was also observed at the cellular level with both a spinning disc confocal and 2-photon microscopy (*Figure 4-c,f*). No fluorescent structures were observed in the CTAP-treated slices examined with the spinning disc or 2-photon microscopes (*Figure 4—figure supplement 1a*). Perhaps surprisingly, NAI-A594 did not efficiently label endogenous DORs in brain slices of the cortex where it has been shown to have DOR immunoreactivity (*Cahill et al., 2001*). Using 2-photon microscopy, sparse labeling of processes was observed that was blocked in CTAP treated slices (*Figure 4—figure supplement 1b*).

## Thalamus and nearby structures

Coronal slices from rat and mouse thalamus were examined next. This brain area exhibits dense MOR binding sites as shown by autoradiography (*Mansour et al., 1988*). Strong fluorescence induced by NAI-A594 was found in the thalamus and habenula (*Figure 5-a,d,e*). Consistent with the results from the striatum, all the labeling was absent in brain slices from MOR knockout rats (*Figure 5-b*). Using a 2-photon microscope, the labeled neurons in the periventricular nucleus of the thalamus (PVT) were identified in a heterogeneous population where other unlabeled neurons were also present (*Figure 5-c*). Fluorescent neurons in the mouse habenula were identified (*Figure 5-d,e,f*), and this labeling was also blocked by CTAP (data not shown).

## Ventral tegmental area and substantia nigra

The labeling of NAI-A594 was next investigated in the substantia nigra and ventral tegmental area of the midbrain. Dopamine cells in these areas are known to express KOR (*Arvidsson et al., 1995b*; *Ford et al., 2007*). There is also a small population of dopamine neurons that are sensitive to ME and thus likely express MOR (*Ford et al., 2006*).

These experiments used THGFP mice to identify dopamine cells. When midbrain slices were incubated in NAI-A594, only ~3% of THGFP

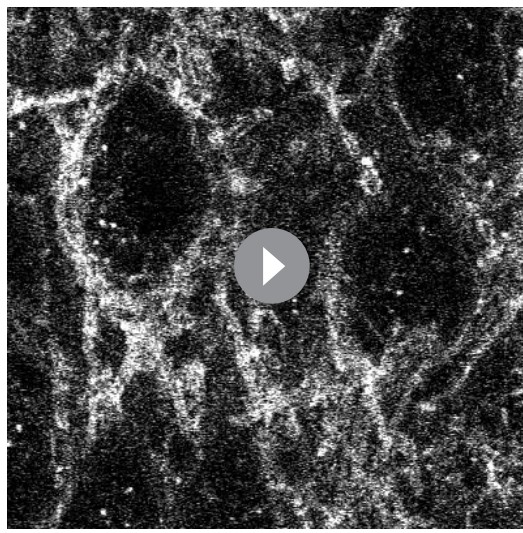

**Video 1.** The image of LC neurons from FMOR mouse labeled with NAI-A594 (30 nM, 1 hr). The movie shows z-sectioning of 2 μm thickness from near surface into 140 μm depth of the brain slice.

DOI: https://doi.org/10.7554/eLife.49319.011

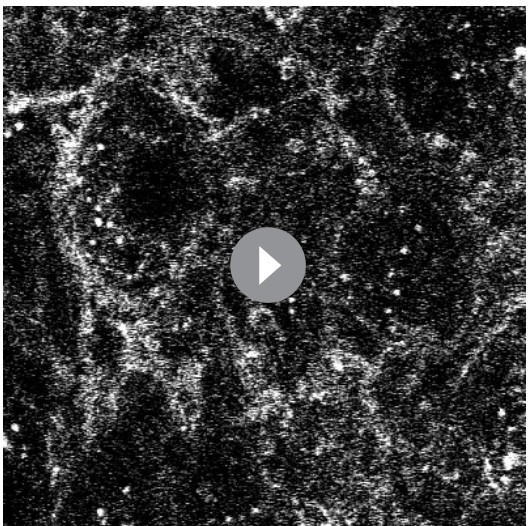

**Video 2.** The image of LC neurons from FMOR mouse labeled with M1-A488 (10 μg/ml, 1 hr). *Videos 1* and *2* were imaged simultaneously with laser excitation at 790 nm.

DOI: https://doi.org/10.7554/eLife.49319.012

**Video 3.** The image of LC neurons from a wild type rat labeled with NAI-A594 (30 nM, 1 hr). The movie shows z-sectioning of 2 μm thickness from near surface into 140 μm depth of the brain slice. The movie was imaged with laser excitation at 810 nm.

DOI: https://doi.org/10.7554/eLife.49319.013

positive cells showed staining with NAI-A594 (*Figure 6A–a,c,c'*; 3 out of 90 cells, eight slices, three animals). Although THGFP negative neurons were labeled with NAI-A594, the population was not quantified (*Figure 6A–a*). Perhaps the most remarkable observation was the amount of Alexa594 fluorescence found on processes lacking GFP fluorescence (*Figure 6A–a,d,d'*). The labeling of these processes was blocked by naloxone (10 μM) and CTAP (1 μM, *Figure 6B*). These results underscore the complexity of opioid systems in the midbrain where inputs of MOR sensitive neurons from several brain areas are reported (*Matsui et al., 2014*). It is expected that NAI-A594 will afford a convenient way to study these neurons and their functionality in slices from wild type animals.

## After NAI-A594 labeling, MORs are functional

Collectively, the results from labeling studies of neurons in brain slices demonstrate that NAI-A594 reacted selectively with the MOR subtype. The next important question is to determine if NAI-A594 labeled neurons function normally. The activation of G protein-coupled inwardly-rectifying potassium conductance (GIRK) in LC neurons was used to determine NAI-A594 labeled MORs remained active. First, does NAI-A594 act as an antagonist? Brain slices containing LC neurons were prepared and whole cell recordings were made. [Met$^5$]enkephalin (ME, 300 nM) caused an outward current that was reduced by half after applying NAI-A594 (500 nM, 5 min, *Figure 7A*, left panel). After washing NAI-A594, the ME-induced current increased toward the pretreatment control indicating that the receptors were free of antagonist. Following this experiment, the slices were removed from the recording chamber and imaged using a

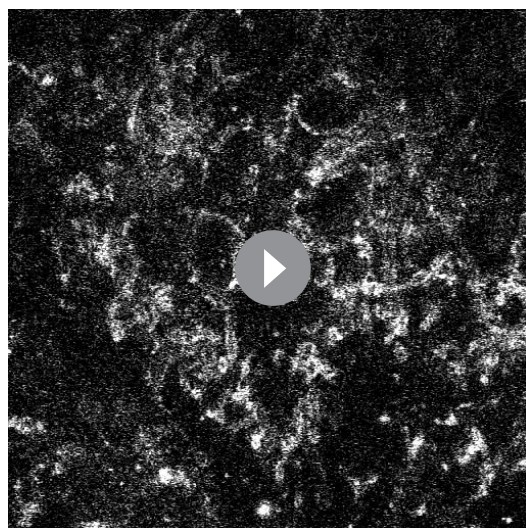

**Video 4.** The image of LC neurons from a wild type rat labeled with M1-A594 (10 μg/ml, 1 hr). The movie was made using the same condition as *Video 3*.

DOI: https://doi.org/10.7554/eLife.49319.014

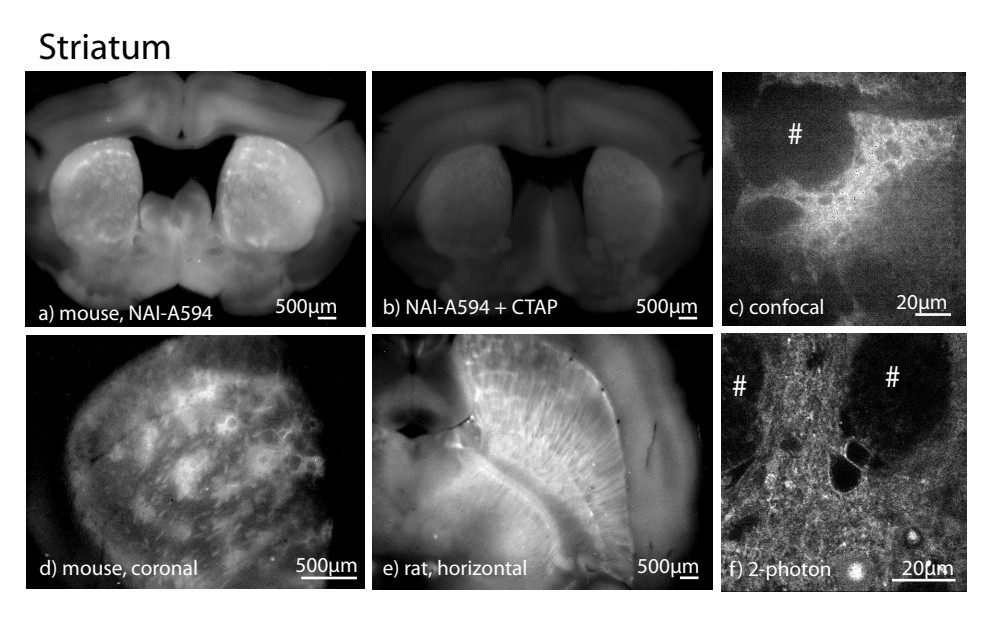

**Figure 4.** NAI-A594 labeled MOR selectively in patches of the striatum in brain slices from wild type animals. All images were prepared in coronal sectioning except as specified. (**a**) NAI-A594 labeled the receptors specifically in the patches of striatum slices from mice. (**b**) The fluorescence was blocked in slices that were treated with CTAP (1 µM). (**c**) A low magnification image taken with a spinning disc confocal microscope showing the fluorescence in a patch of the striatum (20x objective) from a wild type rat. (**d**) A low magnification image of the fluorescence patches in slice from a wild type mouse. (**e**) A low magnification image of the fluorescence patches taken from a horizontal slice of striatum taken from a wild type rat. (**f**) An image taken with a 2-photon microscope of the cells within a fluorescence patch induced by NAI-A594. # is the area of fiber tracts.

DOI: https://doi.org/10.7554/eLife.49319.015

The following figure supplement is available for figure 4:

**Figure supplement 1.** Fluorescent micrographs in striatum and cingulate cortex showed that NAI-A594 primarily labeled endogenous MORs.

DOI: https://doi.org/10.7554/eLife.49319.016

macroscope. A strong fluorescent signal was observed in the LC and parabrachial nucleus (PB, *Figure 7A*, right). The results demonstrate that after the chemical reaction of NAI-A594 with the receptors, the naltrexamine part of this compound was released and dissociated leaving functional receptors.

In general, labeling and identifying MOR containing neurons required only 30–100 nM NAI-A594. To examine the function of the neurons following a typical labeling procedure, whole cell recordings were done with LC slices from wild type rats that had been incubated with NAI-A594 (30 nM, 1 hr). In these experiments, the $EC_{50}$ concentration of ME (300 nM), caused an outward current that was approximately 50% of the current induced by ME at a saturating concentration, 30 µM (*Figure 7B–a*). Labeled LC neurons were also responsive to the $\alpha_2$-adrenoceptor agonist, UK14304 (3 µM). There was no difference in the amplitude of currents induced by ME between labeled neurons and untreated cells in wild type rats (*Figure 7B–b*; *Osborne and Williams, 1995*; *Levitt and Williams, 2012*; *Arttamangkul et al., 2018*). Following these experiments, a post hoc analysis of living neurons was evaluated. In each case recorded cell were filled with Alexa 488 during the whole-cell experiment and later imaged using 2-photon microscopy. The A594-labeled MORs were present on the Alexa488-filled neurons. Other nearby LC neurons and dendrites were also labeled with NAI-A594 (*Figure 7B–c,d*). Thus incubation with a low concentration of NAI-A594 did not affect the concentration response to opioid agonists. When NAI-A594 was used to identify MOR containing neurons in the paraventricular of thalamus areas, all positive staining neurons showed GIRK activation after ME application (n = 4 of 2 slices, data not shown). The results support the advantage of using NAI-A594 to find neurons of interest in mixed populations.

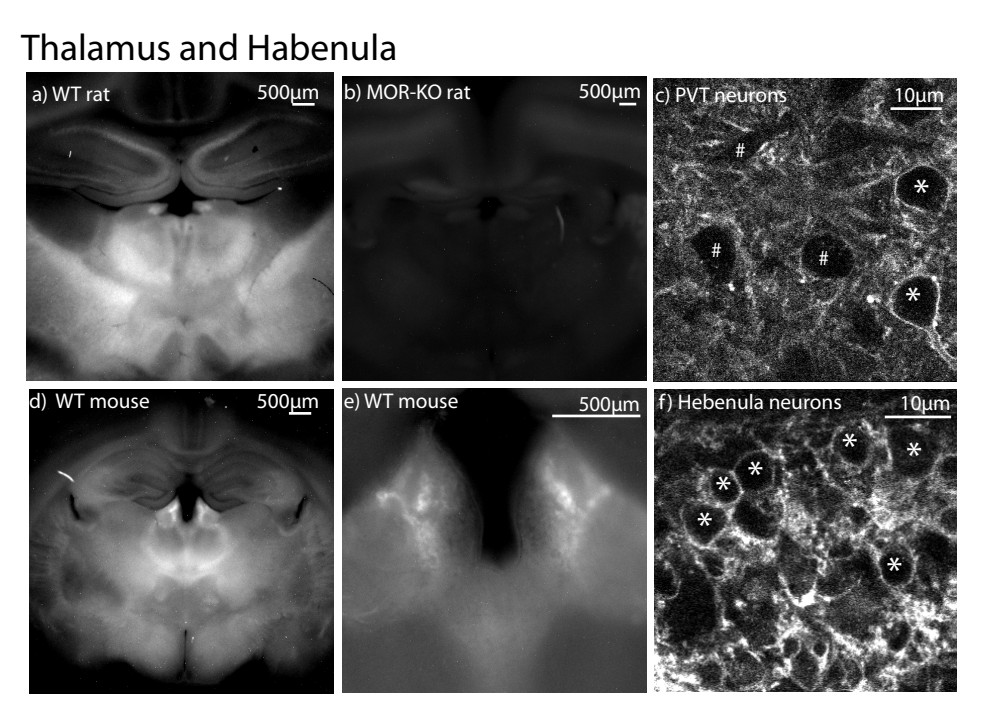

**Figure 5.** Localization of NAI-A594 induced labeling in the thalamus and habenula. (**a**) A low magnification image of a coronal brain slice taken from a wild type rat following incubation with 100 nM NAI-A594 for 1 hr. The habenula and mediodorsal (MD) thalamus show strong fluorescence, (**b**) another slice containing the habenula and MD thalamus taken from a MOR knockout rat, shows no fluorescence. (**c**) An image taken with a 2-photon microscope in the area of the paraventricular thalamus (PVT) showing two cells that were labeled and three cells that were not labeled with NAI-A594. (**d–f**) a sequence of images beginning with low magnification images in the area of the habenula (**d,e**) and ending with a 2-photon image of stained cells in the habenula. *=Labeled and #=non labeled neurons.

DOI: https://doi.org/10.7554/eLife.49319.017

The potential use of NAI-A594 to follow endogenous MOR internalization in brain slices was also studied. NAI-A594 (30 or 100 nM) was incubated in slices of rat LC for 1 hr. Following ME application, the fluorescence of NAI-A594 labeled receptors decreased along the plasma membrane and there were no clear fluorescent puncta in the cytoplasm (*Figure 7C–a,a′*). To increase the labeling efficiency, LC slices were incubated with high concentrations of NAI-A594 (300 nM or 1 µM) for 2 hr. In this condition, agonist-induced receptor internalization was also not observed. To examine whether the labeling of NAI-A594 might interfere with MOR internalization, LC neurons containing FMOR from transgenic mice were used in the next experiments. Co-labeling with M1-A488 and NAI-A594, revealed fluorescence of both dyes on the plasma membrane (*Figure 7C–b,c*). After ME application, fluorescent puncta of M1-A488 were found in the cytoplasm and clearly observed in the dendrites (*Figure 7C–c,c′*). Similar to the results obtained from the rat LC neurons, the fluorescence of NAI-A594 along the plasma membrane decreased without the appearance of fluorescent puncta (*Figure 7C–b,b′*). The results indicated that while A594-linked receptors were not observed in the endosomes of LC neurons, NAI-A594 labeling did not disrupt the internalization of FMORs labeled with M1-A488.

## Discussion

The current study describes a new fluorescent labeling strategy to identify endogenous opioid receptors in living brain tissue. The fluorescent naltrexamine-acylimidazole (NAI) offers a valuable tool for the simultaneous investigation of opioid receptor localization and function. NAI-A594 selectively labeled MORs in brain slices that were visualized with multiple imaging methods. The ability to

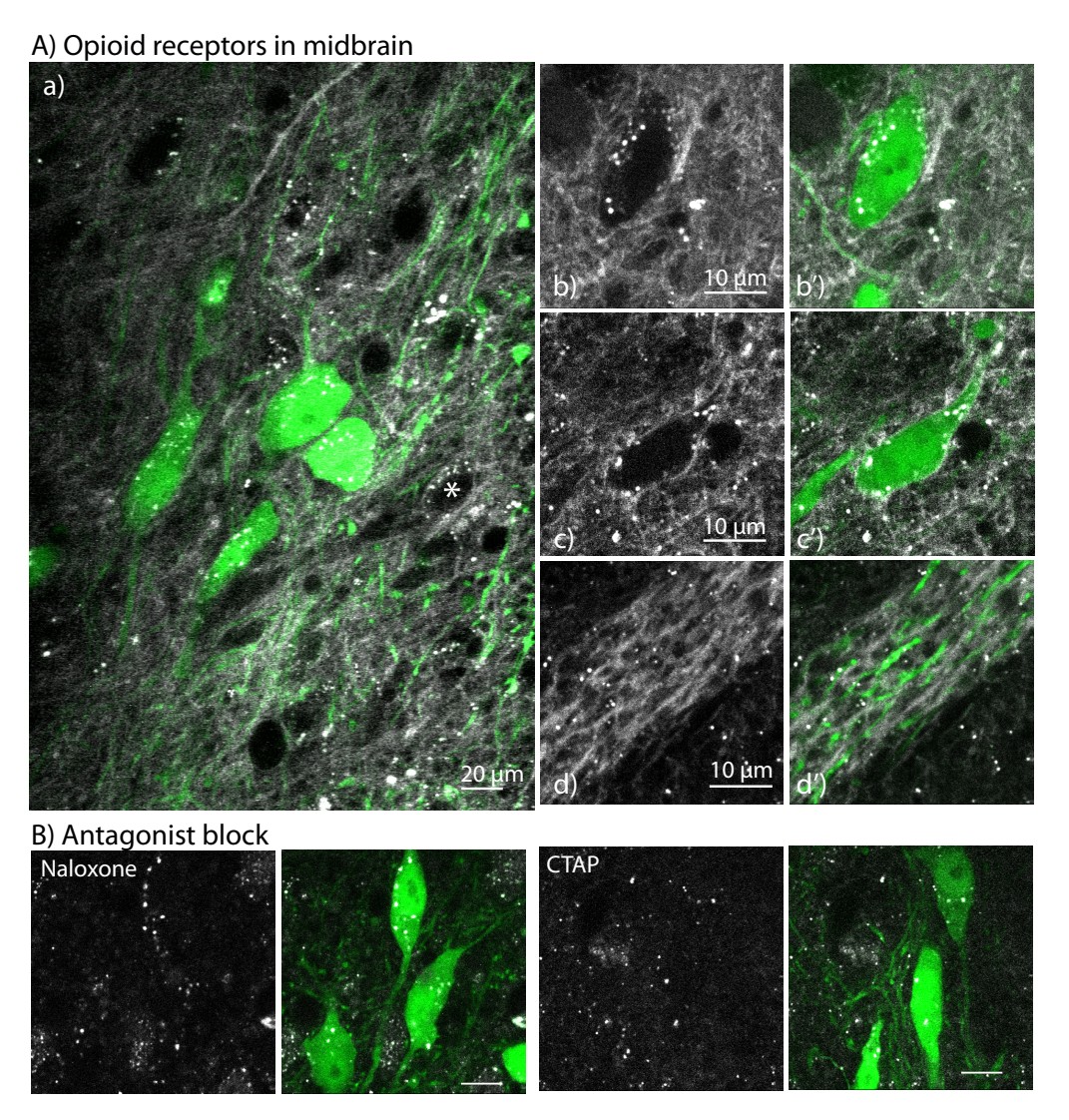

**Figure 6.** The labeling of NAI-A594 in the midbrain of brain slices from TH-GFP mouse. (**A**) Labeling with NAI-A594 identified a small population of MOR neurons (**a**) comparison of the staining induced by NAI-A594 and the GFP fluorescence in TH positive neurons; (**b, b'**) shows one TH-GFP positive (green) neuron without A594 fluorescence along the plasma membrane; (**c,c'**) shows NAI-A594 induced fluorescence on plasma membrane of a GFP positive neuron.In both images, NAI-A594 induced fluorescence was strong in fibers; (**d, d'**) shows the labeling of NAI-A594 on some dendrites that do not originate from green neurons. (**B**) Incubation of slices with naloxone (10 μM) or CTAP (1 μM) completely blocked the NAI-A594 induced staining. *=An A594-labeled neuron that was TH negative.

DOI: https://doi.org/10.7554/eLife.49319.018

identify neurons expressing endogenous MORs in living brain slices from wild type rats and mice highlights the benefit of this reagent for a wide array of investigations that will advance the understanding of the opioid system.

## Chemical labeling of MORs

This study shows that NAI-A594 can be used in low nanomolar concentrations to label endogenous MORs, thus providing the advantage of negligible non-specific background fluorescence and the absence of residual antagonism of NAI-A594 and its product. Incubation times of live brain slices can be varied from minutes to hours at near physiological temperatures. It should be noted that an application of NAI-A594 at a high concentration of 500 nM took only 5 min to stain MORs of LC and PB neurons in the brain slice thus indicating that NAI-A594 can efficiently label endogenous MOR.

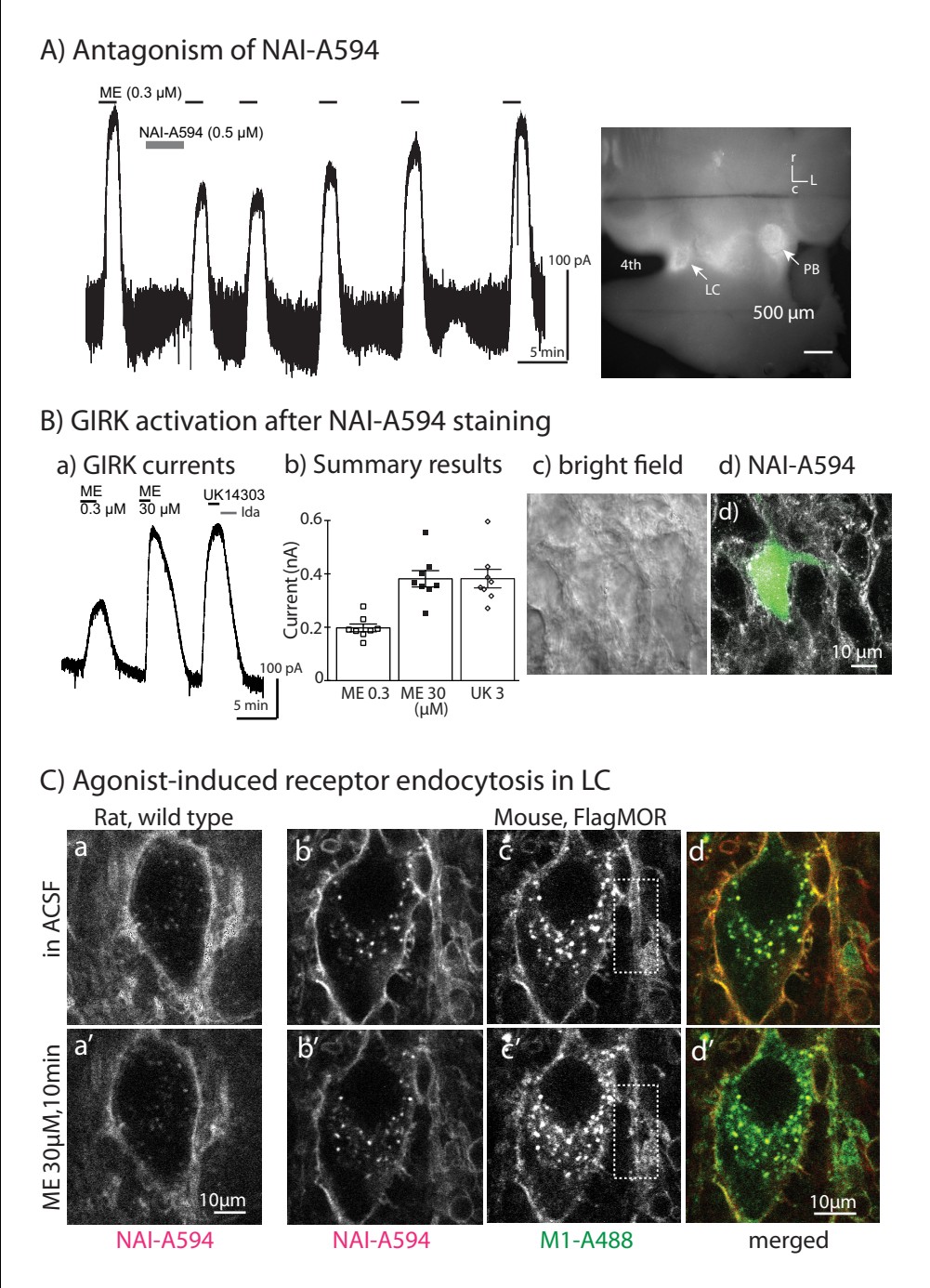

**Figure 7.** Labeled MORs are functional. Whole-cell recordings were used to measure GIRK activation induced by ME. (**A**) Application of NAI-A594 (500 nM) by circulating for 5 min caused a decrease of ME-activated GIRK current. The reduction was reversible after washing out of NAI-A594. The image on the right shows fluorescent staining of the same slice following the recording. The strong fluorescence in the areas of LC and parabrachial nucleus (PB) was observed. (**B–a**) The tracing shows GIRK activation of a LC neuron from the slice that was incubated in NAI-A594 (30 nM, 1 hr), (**B–b**). Summarized results show amplitude of the currents induced by the EC$_{50}$ concentration of ME (300 nM) and the saturating concentration of ME (30 µM) or UK14303 (3 µM). (**B–c,d**) The micrographs demonstrate the bright field image (**c**) and the corresponding fluorescent image of a recording neuron labeled with NAI-A594 that was filled with A488 (**d**). (**C**) 2-Photon images of LC neurons from a wild type rat labeled with NAI-A594 show fluorescence along the plasma membrane before (**a**) and after 30 µM ME for 10 min (**a'**). The 2-photon images of LC neurons from an FMOR transgenic mouse labeled with M1-A488 and NAI-

*Figure 7 continued on next page*

*Figure 7 continued*

A594 show fluorescence staining before (**b,c,d**) and after 30 μM ME for 10 min (**b',c',d'**). The dyes were simultaneously excited at 790 nm and autofluorescence was notably observed in the cell somata. White boxes illustrated the fluorescent staining of dendrite with M1-A488 before (**c**) and after ME application (**c'**).

DOI: https://doi.org/10.7554/eLife.49319.019

The following figure supplement is available for figure 7:

**Figure supplement 1.** Tracing from whole cell recording showing the activation of GIRK by ME (300 nM) was blocked by application of NAI-AK (300 nM).

DOI: https://doi.org/10.7554/eLife.49319.020

The weak biological antagonism of NAI-A594 suggests that the compound itself has a low affinity as shown in the binding assays and can be competitively displaced by opioid agonists. Labeling brain slices using low concentrations of NAI-A594 or NAI-A488 can be achieved within an hour. Interestingly, the labeling of purified receptors with NAI-A488 was slower when compared to the labeling of receptors on the living cells. One possible factor is the incubation temperature, which is done at room temperature to stabilize the purified receptors. In addition, the purified receptors may not as frequently form proper conformations for β-naltrexamine to bind and efficiently guide the reaction.

## NAI-A594 labels MOR in brain slices

Here we demonstrate that NAI-A594 primarily and covalently labels MOR on neurons in brain slices. This is unexpected given that naltrexamine binds to all three opioid subtypes at low nanomolar concentrations (*Li et al., 2009*). The design of NAI to have a long linker between the acylimidazole and the amine of naltrexamine (10 atoms apart) was expected to transfer the fluorophore to distant amino acids away from the binding sites. The side chains of Ser, Thr, Tyr or Lys on the second (EL2) or third (EL3) extracellular loops of the receptor are hypothetical sites for modification (*Figure 1— figure supplement 1*). The ability of fluorescent NAI to predominantly modify MOR likely relies on several factors, including the availability of reactive amino acid side chains and affinity of fluorescent NAI for the different opioid receptors. For example, MOR has 12 amino acids that are potentially capable of reacting with fluorescent NAI, while KOR has 8 and DOR only 3. The paucity of available reactive amino acids in DOR may be one factor contributing to lack of A594 modification in brain slices, despite similar affinities of NAI-A594 for MOR and DOR. Conversely, KOR may have more potential modification sites compared to DOR, but has the lowest affinity for NAI-A594 compared to MOR and DOR (*Table 1*).

The selectivity of NAI-A594 for MOR over DOR in brain slice experiments contrasts with the experiments in HEK293 cells where the labeling of FMOR and FDOR were comparable. It is possible that the density of DOR on the plasma membrane in vivo is lower compared to MOR (*Wang et al., 2008* and references therein). Importantly, the Flag epitope-tagged receptors used in the HEK cells contains a signal peptide to enhance endoplasmic reticulum membrane insertion and plasma membrane expression. The development of compound that is capable of labeling intracellular receptors will be advantage. Changing the fluorescent dye from a hydrophilic and charged Alexa series to one that is smaller and not charged may increase membrane permeability of NAI compounds.

The labeling of MORs with NAI is different from the genetically expressed DOR-GFP and MOR-RFP in knockin mice (*Scherrer et al., 2006* and *Erbs et al., 2015*). Of course the genetically expressed receptors are found both on the plasma membrane and within the cytoplasm in various states of development. Labeling of receptors with NAI-A594 is only plasma membrane associated. The primary advantage of the NAI-A594 is that it is also working with other species particularly rats. The ability to study endogenous receptors in wild type animals of any species offers studies that were otherwise limited to post hoc analysis with antibody labeling.

## The labeled neurons are functional

One key characteristic for a traceless chemical label is that the receptor function is not altered. We confirmed that NAI-A594 was an antagonist at MOR. Pre-application of NAI-A594 (500 nM) decreased receptor activation by nearly half while there was less than 10% inhibition by NAI-A594 at lower concentrations (30–100 nM, data not shown). Like other fluorescent small-molecule ligands,

the weak antagonism of NAI-A594 is probably due to the bulkiness of Alexa594 moiety because NAI-AK, the reactive probe without the dye could significantly block MOR activation when used at 300 nM (*Figure 7—figure supplement 1*). The low antagonism of NAI-A594 is advantageous particularly when only 30–100 nM of NAI-A594 is sufficient for labeling the receptors. In this case, the antagonistic property of NAI-A594 will be minimal and the residual of NAI-A594 or the liberated-naltrexamine product will be removed in a short time. Consistent with this idea, LC neurons labeled with NAI-A594 are able to activate GIRK in response to ME.

In addition to GIRK activation, further evidence that the labeled MOR receptors are functional comes from the agonist-induced internalization studies. Intracellular A594-puncta were observed upon exposure to ME in HEK293 cells expressing FMORs double labeled with M1-488 and NAI-A594 thus indicating that A594-labeled MORs can internalize in response to agonists. Interestingly, when similar internalization experiments are performed on M1-A488/A594 labeled MORs in LC slices, internalized signals appear only from M1-A488. In contrast, there is a noticeable decrease in A594 fluorescence signal on the plasma membrane. A quenching phenomenon of A594 in the acidic environment of endosomes can be ruled out because the fluorescence of A594 is pH insensitive (*Panchuk-Voloshina et al., 1999*). Surprisingly, similar decline in fluorescence intensities of 30–40% was observed when either ME, morphine or naloxone were applied onto the labeled LC neurons (*Figure 3—figure supplement 1A*). Because morphine is a non-internalizing agonist and naloxone is an antagonist, the results suggest that the decrease of A594 fluorescence probably occurs on the plasma membrane. This portion of labeling is competitive as also shown in HEK293 cells. Flow cytometry results also showed that a portion of labeling was displaced by naloxone (*Figure 2B*). The competitive labeling occurs when NAI-A594 occupies the binding site even on receptors that have been previously covalently linked with the fluorescent ligand.

Another potential explanation for the fluorescence decrease may be due to a quenching event caused by nearby amino acids such as tryptophan, tyrosine, histidine and methionine as a result from receptor-conformation change after drug application (*Mansoor et al., 2010*). However this phenomenon is expected to be reversible after washing out of the drugs but the decrease in fluorescence observed experimentally was not. The decrease in fluorescence intensity after application of drugs might limit visualization of A-594 labeled receptors in the endosomes. In addition, many neurons often show autofluorescence at the excitation wavelengths suitable for A488 and A594, thus further obscuring the detection of low fluorescence of A594 labeled receptors in the endosomes.

## Conclusions

The selectivity of NAI-A594 for the MOR makes this reagent a valuable tool to study the functionality and circuitry of MORs in many brain areas. The 'traceless affinity labeling' introduced by Hamachi and his colleague is applicable for many membrane proteins including G protein-coupled receptors as shown in this study. The advantage of this ligand-guided labeling is the ability to detect endogenous receptors in living slices. The labeling process is very simple and can be done in any buffer, at room or physiological temperatures and over reasonable time course.

## Materials and methods

**Key resources table**

| Reagent type (species) or resource | Designation | Source or reference | Identifiers | Additional information |
|---|---|---|---|---|
| Strain (mouse, C57BL/6J) | wildtype | Jackson Laboratories | Stock # 000664 RRID: IMSR_JAX:000664 | |
| Strain (rat, Sprague Dawley) | wildtype | Charles River Laboratories | Stock # 001 RRID: RGD_734476 | |
| Genetic reagent (mouse) | C57BL/6J (THGFP) | *Sawamoto et al., 2001* | N/A | Dr. Kazuto Kobayashi, Fukushima Medical University |

*Continued on next page*

*Continued*

| Reagent type (species) or resource | Designation | Source or reference | Identifiers | Additional information |
|---|---|---|---|---|
| Genetic reagent (mouse) | C57BL/6J (FLAGMOR) | *Arttamangkul et al., 2008* | N/A | Dr. John T. Williams, OHSU |
| Genetic reagent (mouse) | C57BL/6J (MOR knockout) | *Schuller et al., 1999* | *Oprm1*-exon-1 knockout mice | Dr. John E. Pintar, RWJMS |
| Genetic reagent (rat) | Sprague Dawley (MOR knockout) | Horizon Discovery | *Oprm1*-exon-2 knockout rat | |
| Antibody | Anti-Flag-M1 (mouse monoclonal) | Sigma | Cat# F-3040; RRID: AB_439712 | Conjugation with Alexa488 or Alexa594 and used at 3 or 10 µg/ml |
| Antibody | Anti-MOR (UMB3-rabbit monoclonal) | ABCAM | Cat# ab134054 | 1:100 |
| Antibody | Anti-Alexa Fluor 488 (rabbit polyclonal) | Thermo Fisher | Cat# A-11094; RRID: AB_221544 | 1:500 |
| Antibody | Anti-Flag M1 agarose affinity gel | Sigma | Cat# A4596 | 50 µl/ml |
| Chemical compound, drug | β-naltrexamine | NIDA | N/A | Dr. Kenner Rice, NIDA |
| Chemical compound, drug | [Met$^5$]enkephalin | Sigma | Cat# M6638 | |
| Chemical compound, drug | Morphine | NADA | N/A | |
| Chemical compound, drug | CTAP | Sigma | Cat# C6352 | |
| Chemical compound, drug | Naloxone HCl | Hello Bio | Cat# HB2451 | |
| Chemical compound, drug | AFdye 594 azide | Click Chemistry Tools | Cat# 1295–1 | |
| Chemical compound, drug | AFdye 488 azide | Click Chemistry Tools | Cat# 1275–1 | |
| Chemical compound, drug | NAI-A594 | This manuscript | N/A | Dr. John T. Williams, OHSU |
| Chemical compound, drug | NAI-A488 | This manuscript | N/A | Dr. John T. Williams, OHSU |
| Software | ScanImage | Vidrio Technologies, LLC | ScanImage, RRID: SCR_014307 | |
| Software | Prism | GraphPad | 6.0d for Mac OS X | |
| Software | ImageJ | www.imagej.nih.gov | 1.44o | |

## Drugs and chemicals

Naloxone and MK-801 were purchased from Hello Bio (Princeton, NJ). [Met$^5$]enkephalin and CTAP were purchased from Sigma (St. Louis, MO). UK14304 tartrate and idazoxan were from Tocris (R and D system, Minneapolis, MN). Morphine alkaloid was obtained from National Institute on Drug Abuse, Neuroscience Center (Bethesda, MD) and was converted to HCl salt with 0.1 M HCl as a stock solution. All drugs were diluted to the tested concentrations in artificial cerebrospinal fluid (ACSF) and applied during superfusion. All salts used in electrophysiological experiments were purchased from Sigma.

## Animals

All animal uses were conducted in accordance with the National Institutes of Health guidelines and with approval from the Institutional Animal Care and Use Committee of OHSU. Animals were housed as a group (2–3 rats or 2–5 mice) to a cage in the animal care facility on a 12-hr light/dark cycle.

Food and water were available ad libitum. Juvenile rats (21–28 days) both male and female Sprague-Dawley rats were raised from two breeding pairs that were purchased from Charles River Laboratories (Wilmington, MA). MOR-knockout rats (MOR-KO, Sprague-Dawley) were obtained as breeding pairs from Horizon Discovery (St. Louis, MO) and were bred in house. The *Oprm1* gene was inactivated with ZFN target site within exon-2 (*Arttamangkul et al., 2018*). Both male and female MOR-KO rats were used at the same age as wild type rats. Juvenile mice (21–28 days), both male and female C57BL/6J background were used for all genotypes. Wild type mice were raised from breeders obtained from the Jackson Laboratory (Bar Harbor, ME). MOR-KO mice were gifted from Dr. John Pintar (RWJMS). These mice had *Oprm1* gene disrupted in exon-1. Transgenic FLAGMOR mice were generated as described (*Arttamangkul et al., 2008*). The FLAG epitope was targeted to the N-terminus of murine MOR1 and inserted to the rat tyrosine hydroxylase (*Th*) gene. Hemizygous animals (FLAGMOR-*Tg*/+) were used in the experiments. Transgenic THGFP mice were received from Dr. Kobayashi (*Sawamoto et al., 2001*). The THGFP mice have GFP inserted to the rat tyrosine hydroxylase (*Th*) gene. The line was maintained as hemizygous.

## Chemical labeling of MOR in HEK293 cells

HEK293 cells stably expressing FLAG epitope-tagged MOR (FMOR, gifted from Dr. Mark von Zastrow, UCSF) and FLAG epitope-tagged DOR (FDOR, gifted from Dr. Manojkumar Puthenveedu, U of Michigan) were grown and maintained in Dulbecco's minimal essential media (DMEM, Gibco, Grand Island, NY) containing 10% fetal bovine serum (FBS), Geneticin sulfate (0.5 mg/ml, ThermoFisher, Waltham, MA) and antibiotic-antimycotic (1x, ThermoFisher). FLAG epitope-tagged KOR (FKOR, the plasmid was obtained from Dr. Mark von Zastrow, UCSF) was transiently transfected into HEK293 cells (ATCC authenticated lines CRL-1573) using Lipofectamine 2000 (ThermoFisher, Waltham, MA). Cell line cultures were free of mycoplasma contamination. Cells were plated onto a cover glass (poly-L-lysine coated, 12 mm, #1 thick, NeuVitro, Vancouver, WA) and used the next day. Cells were incubated in NAI-A594 or NTX-594 (30 nM in cell growth media, 30 min) following with anti-Flag (M1)-A488 (3 µg/ml, 5 min) in the incubator (37°C, 5% $CO_2$). The cover glass was placed in the imaging chamber and cells were continually perfused with artificial cerebrospinal fluid (ACSF) solution containing (in mM) 126 NaCl, 2.5 KCl, 1.2 MgCl2, 2.6 CaCl2, 1.2 NaH2PO4, 11 D-glucose and 21.4 NaHCO3 (95%O2/5%CO2). Experiments to compare labeling of NAI-A594 and NTX-A594 were done at room temperature. Naloxone (1 µM) was superfused for 10 min. Internalization experiments were done at 30°C and with application of [Met$^5$]enkephalin (ME, 10 µM) by perfusion for 7–10 min. Imaging was performed as previously described (*Birdsong et al., 2013*) with an upright spinning disc confocal microscope and a 60x water immersion lens (Olympus LUMIPLAN, NA 1.0). Alexa488 dye and Alexa594 were excited at 488 nm and 561 nm lasers, respectively.

## Flow cytometry

HEK293 cells expressing FMOR were labeled with M1-A594 (3 µg/ml) in buffer A [DMEM containing 5% fetal bovine serum (FBS) and no phenol red] and incubated on ice bath for 1 hr. The conjugated M1-A594 was spectrometrically measured and calculated to have an average of 4 dye molecules per one molecule of IgG. Labeling of cells with NAI-A594 or Natrexamine-A594 (NTX-A594) was done in buffer A and incubated in an incubator at 37°C and 5% $CO_2$ for 1 hr. All samples were washed three times with DPBS (2 ml, Gibco) and resuspended in ice-cold buffer A (0.3 ml). In some samples, 10 µM naloxone in buffer A was used for washing. Flow cytometry measurements were done using Fortessa (BD Bioscience, Singapore) at the core facility of Oregon Health and Science University (OHSU, Portland, OR). 5000 to 10,000 cells were counted for each condition. Experiments were repeated three times for each condition.

## FMOR immunoprecipitation-Western blots

To confirm that NAI-A488 labeling of FMOR expressing cells was specific, lysates were prepared from cells lacking FMOR, from unlabeled FMOR expressing cells, from cells labeled in the presence of naloxone (10 µM), and from cells labeled in the absence of any other treatments. Cell lysates were produced by nutating cells in lysis buffer (20 mM Tris pH 7.4, 100 mM NaCl, 1.0% DDM, 0.5 mM PMSF, 5 ug/ml leupeptin, 1X Halt protease inhibitor cocktail) for 1 hr, at 4°C, followed by centrifugation at 100,000 x g for 30 min and collecting the resulting supernatant. FMOR was

immunoprecipitated from cell extracts using a rabbit anti-Mu opioid receptor antibody (Abcam, Cambridge, UK, used at a dilution of 1:100), and 30 µl of a 50% slurry of Protein A Sepharose (GE Healthcare, Marlborough, MA). Approximately 170 µg total protein was immunoprecipitated per each condition. The immunoprecipitated proteins were denatured in sample buffer/0.1 M dithiothreitol (DTT) then split into two aliquots and run on SDS-PAGE, electro-transferred to PVDF membranes, and probed with primary antibodies, either M1 anti-FLAG (Sigma, 1:500 dilution) or rabbit anti-Alexa-488 (ThermoFisher, 1:500 dilution), followed by washing and incubation with secondary antibodies, either goat-anti mouse Alexa-680 (ThermoFisher Scientific, A12058 1:5000 dilution) or goat-anti rabbit Alexa-680 (ThermoFisher Scientific, A12076, 1:5000 dilution). The protein bands were visualized using a LICOR Infrared Imager and Odyssey V3.0 application software. All antibody and wash solutions for blots that were probed with the M1 Mab were supplemented with $CaCl_2$ (1 mM).

## Fluorometry of solubilized cell extracts

The fluorescence of cell lysates was measured using a fluorimeter (PTI QuantaMaster) with excitation wavelengths (2 nm slit settings) of 470 nm or 590 nm for Alexa-488 or Alexa-594 conjugated compounds, respectively. Fluorescence emission (12 nm slit settings) spectra were collected from 485 to 600 nm (Alexa-488) or from 600 to 700 nm (Alexa-594) using a scan rate of 1 nm/s. The fluorescence emission data were buffer subtracted, normalized to maximal fluorescence in the labeled sample and plotted versus wavelength.

## FMOR purification

Pellets from FMOR expressing HEK293 cells were solubilized in solubilization buffer [20 mM HEPES pH 7.5, 150 mM NaCl, 30% glycerol, 0.5% n-dodecyl-β-D-maltoside (DDM), 0.3% (3-((3-cholamido-propyl) dimethylammnonio)−1-propanesulfonate (CHAPS), 0.03% cholesteryl hemisuccinate-tris salt (CHS), naloxone (1 µM), Leupeptin (5 µg/ml), PMSF (0.5 mM), 1X Halt protease inhibitor cocktail (ThermoFisher Scientific, 87786, Waltham, MA)], nutated for 1 hr at 4°C, $CaCl_2$ added to a final concentration of 1 mM, and then centrifuged at 100,000 x g for 1 hr, at 4°C. The resulting supernatant was then applied to an M1 anti-Flag column (Sigma, A4596, St. Louis, MO). 50 µl of M1 bead slurry was used for each ml of clarified lysate. The column was capped and the bead-lysate mixture was then nutated overnight at 4°C. The next morning the column was attached to a ring stand and the beads allowed to settle. The beads were then washed 4X in washing buffer (HEPES, 20 mM pH 7.5, NaCl 150 mM, $CaCl_2$ 1 mM, 0.1% DDM, 0.06% CHAPS, 0.006% CHS), followed by elution in washing buffer that lacked $CaCl_2$ and contained EDTA 2 mM. Eluted FMOR concentrations were determined by UV-Vis spectroscopy (Shimadzu, Kyoto, Japan) using an extinction coefficient, which we estimated from the protein sequence (ExPASy ProtParam tool), for 280 nm absorbance of 62,340 $M^{-1}$ $cm^{-1}$. Purity was assessed by SDS-PAGE followed by protein staining with InstantBlue (Expedeon, ISB1L, Sandiago, CA).

## NAI-A488 FMOR binding assay

Purified FMOR protein was labeled with NAI-A488 at room temperature (RT, 20°C) by incubating FMOR (0.75 µM) with NAI-A488 (4.7 µM) in the absence or presence of naloxone (100 µM) in a final volume of 159 µl. At 0, 4, 24, 48, and 96 hr, 30 µl aliquots were collected and denatured in sample buffer plus 0.1 M DTT. The samples were then run on 10% SDS-PAGE gels. Before incubating gels with InstantBlue protein stain, fluorescent bands were imaged using an Alpha Innotech FluorChem 5500 gel documentation system, using the SYBR green channel. Protein band intensities were determined using Image J.

## Chemical labeling of MOR in brain slices

Rats and mice were anesthetized with isoflurane and killed. The brain was removed and placed in warm (30°C) oxygenated ACSF plus (+)MK-801 (1 µM). Brain slices were prepared (250 µm) using a vibratome (Leica, Nussloch, Germany) and incubated in oxygenated warm (34°C) ACSF containing (+)MK-801 (10 µM) for 30 min before use. Slices were incubated in oxygenated ACSF containing NAI-A594 or NAI-A488 (10–100 nM) for 1 hr at 30°C. Labeled slices were transferred to an upright microscope (Olympus BX51W1, Center Valley, PA) equipped with a custom-built 2-photon apparatus

and a 60x water immersion lens (Olympus LUMFI, NA 1.1). The dye was excited at 810 nm for Alexa594 and Alexa568, and 790 nm for Alexa488 and Alexa594 at the same time. Data were acquired and collected using ScanImage Software (*Pologruto et al., 2003*). Slices were submerged in continuous flow of ACSF at a rate of 1.5 ml/min and drugs were applied via superfusion. All experiments were done at 35°C. Macroscopic images of labeled slices were captured with a Macro Zoom Olympus MVX10 microscope and a MV PLAPO2xC, NA 0.5 lens (Olympus). Alexa594 dye was excited with yellow LED (567 nm). The slices were kept under ACSF in a petri dish during data acquisition. Data were acquired using Q Capture Pro software (Q Imaging, British Columbia, Canada). In the other experiments, a spinning disc confocal microscope and a 20x water immersion lens (Olympus UMPlanFL NA 0.95) was used to take pictures of the labeled brain slices. Alexa488 dye was excited by a 488 nm laser whereas Alexa568 and Alexa594 dyes were excited by 561 nm laser. The slices were maintained alive under superfusion of oxygenated ACSF (30°C).

## Electrophysiology

Whole-cell recordings were done with an Axopatch-1D amplifier in voltage-clamp mode ($V_{hold}$ = −60 mV). Recording pipettes (1.7–2.1 MΩ, TW150F-3, World Precision Instruments, Saratosa, FL) were filled with internal solution containing (in mM): 115 potassium methanesulfonate or potassium methyl sulfate, 20 KCl, 1.5 MgCl2, 5 HEPES potassium salt, 2 BAPTA, 2 Mg-ATP 0.2 Na-GTP, pH 7.4, 275–280 mOsM. Series resistance was monitored without compensation and accepted for further analysis when < 15 MΩ. Data were collected at 400 Hz with PowerLab (Chart version 5.4.2, AD Instruments, Colorado Springs, CO). [Met[5]]enkephalin was applied by bath superfusion. Bestatin (10 µM, Sigma) and thiorphan (1 µM, Sigma) were included in enkephalin solution to prevent enzymatic degradation. The alpha-2-adrenoceptor agonist, UK14304 (3 µM) and the antagonist Idazoxan (1 µM) were used as controls.

## Radioactive binding assays

Radioligand binding assays were performed as described (*Arttamangkul et al., 1997*). Cloned rat mu opioid receptor (MOR) and kappa opioid receptor (KOR), and mouse delta opioid receptor (DOR) stably expressed separately in Chinese hamster ovary (CHO) cells were grown and used to make membrane preparations. [[3]H]DAMGO, [[3]H]diprenorphine and [[3]H]DPDPE were used as radioligands for MOR, KOR and DOR, respectively. Incubations were carried out in triplicate with varying concentrations of NAI-A594 ($0.1-10 \times 10^3$ nM) for 90 min at room temperature. DAMGO, Dyn A(1-13)NH$_2$ and DPDPE were used as control ligands. Nonspecific binding was determined in the presence of 10 µM unlabeled DAMGO, Dyn A(1-13)NH$_2$, and DPDPE for MOR, KOR and DOR, respectively.

## Data analysis

Data were graphed using GraphPad Prism software (La Jolla, CA). All images were processed and analyzed using Image J (NIH, Bethesda, MD).

## Synthesis of NAI-AK

The synthesis including reagents and chemical characterization of NAI-A594 are described in detail below. β-Naltrexamine was synthesized and purified as described previously (*Sayre and Portoghese, 1980*).

**Chemical structure 1.** Synthesis of compound 1: 3-{[2-(1H-imidazol-4-yl)ethyl]carbamoyl}propanoic acid.
DOI: https://doi.org/10.7554/eLife.49319.021

Succinic anhydride (1.76 g, 17.6 mmol) was dissolved in 15 mL of anhydrous DMF, under nitrogen atmosphere. A solution of histamine −2 HCl (2.93 g, 15.9 mmol) in 25 mL anhydrous DMF and triethylamine (31.8 mmol, 4.43 mL) was added dropwise and stirred. The reaction mixture was carried on overnight at room temperature. The precipitate was filtered and the filtrate was concentrated under vacuum. Ethanol (20 mL) was added to the concentrated solution and the mixture was chilled

at 4°C overnight. The white precipitate was filtered, washed with cold ethanol, and dried in vacuo. 2.83 g of a white solid was obtained (84%). 1H NMR (D₂O): δ (ppm) 8.5 (s, 1H), 7.20 (s, 1H), 3.41 (t, J = 6.37 Hz, 2H), 2.86 (t, J = 6.37 Hz, 2H), 2.35 (s, 4H).

**Chemical structure 2.** Synthesis of compound 2: N-[(1S,5R,13R,14R,17S)—4-(cyclopropylmethyl)—10,17-dihydroxy-12-0xa-4-azapentacyclo[9.6.1.0^{1,13}.0^{5,17}.0^{7,18}]octadeca-7,9,11(18)-trien-14-yl]-N'-[2-(1H-imidazol-4-yl) ethylbutanediamide.
DOI: https://doi.org/10.7554/eLife.49319.022

Naltrexamine 2HCl (9.97 mg, 24 μmol) was dissolved in DMF to give 16 mM solution. The naltrexamine solution (1.5 mL) was transferred to a 2.5 mL microwave vial, following with compound 1 (20.3 mg, 96 μmole), HBTU (72 μmole, 27.3 mg), triethylamine (two drop,~40 uL). Additional of 1 mL DMF was added to wash down the reagents on the side of vial. Microwave reactor was set to 150° C, 10 min, normal absorption. HPLC analysis of the crude material using a gradient solution of 5% to 65% water and acetonitrile containing 0.1% trifluoroacetic acid in 30 min and a reverse phase (Jupiter, 5 μm, C18, 300 Å, 250 × 4.6 mm) confirmed the reaction was completed and there was no naltrexamine in the crude. The crude was purified using silica gel chromatography (DCM/MeOH/NH4OH 4/1/1). The eluent was collected and evaporated. The electron-spray ionized mass spectrum confirmed the product peak at 536.5 (m/z+1). Yields 12 mg (94%). 1HNMR (400 MHz, CD₃OD): δ (ppm) 8.76 (s, 1H), 7.36 (s, 1H), 7.05 (d, J = 8.4 Hz, 1H), 6.90 (d, J = 8.8 Hz, 1H), 4.68 (d, J = 8 Hz, 1H), 4.37 (m, 2H), 4.22 (s, 2H), 3.98 (m, 1H), 3.80 (t, J = 4 Hz, 2H), 3.61 (m, 1H), 3.14 (m, 1H), 2.92 (m, 3H), 2.66 (m, 3H), 2.55 (m, 2H), 2.44 (m, 2 hr), 1.90 (m, 1H), 1.67 (m, 3H), 1.12 (m, 1H), 0.85 (m, 1H), 0.76 (m, 1H), 0.53 (m, 2H). MS (ESI pos) m/z found 536.5 (M + H).

**Chemical structure 3.** Synthesis of compound 3: 2,5-dioxopyrrolidin-1-yl 2-(pro-2-yn-1-yloxy)ethyl carbonate.
DOI: https://doi.org/10.7554/eLife.49319.023

A solution of 2-(Prop-2-yn-1-yloxy)ethanol (25.0 mg, 0.25 mmol), N,N'-disuccinimidyl carbonate (96 mg, 0.375 mmol), and triethylamine (140 uL, 1 mmol) in CH₃CN (3 mL) was stirred for 2 hr at 40° C. The solvent was evaporated and product was purified using silica gel chromatography (0% to 50% ethyl acetate/hexane). The product was detected by vanillin stain. The eluent was collected and evaporated giving the product of clear oil. Yielded 48 mg (80%). 1HNMR (400 MHz, CDCl₃): δ (ppm) 4.56 (m, 2H), 4.19 (d, J = 2.38 Hz, 2H), 3.80 (m, 2H), 2.81 (s, 4H), 2.52 (t, J = 2.38 Hz, 1 hr).

**Chemical structure 4.** Synthesis of compound 4: 2-(prop-2-yn-1-yloxy)ethyl 4-[2-(3-{[1 s,5R,13R,14R)—4-(cyclopropylmethyl)—10,17-dihydroxy-12-oxa-4-azapentacyclo[9.6.1.0^{1,13}.0^{5,17}.0^{7,18}]octadeca-7,9,11(18)-trien-14-yl]

carbamoyl}propanamido)ethyl]—1H-imidazol-1-carboxylate.
DOI: https://doi.org/10.7554/eLife.49319.024

A solution of compound 3 (48 mg, 200 µmol) in 2 mL DMF was combined with compound 2 (~24 µmol) in a vial. Pyridine five drops were added, and the reaction mixture was stirred at room temperature overnight. Then the reaction mixture was filtered over celite. The filtrate was concentrated and azeotroped with toluene. The residue was washed twice with ether. Ethyl acetate was added to give a solid product. Filtrate of excess compound 3 was removed. The solid product was dissolved in methanol and filtered over celite. Methanol was then evalprated under vacuo and light brown solid powder was obtained. Yields 4.5 mg (66%). The electron-spray ionized mass spectrum confirmed the product peak at 662.6 (m/z+1). 1HNMR (400 MHz, CD3OD): δ (ppm) 8.22 (s, 1H), 7.40 (s, 1H), 6.74 (d, J = 2.12 Hz, 1H), 4.57 (m, 3H), 4.24 (d, J = 2.4 Hz, 2H), 3.90 (m, 3H), 3.63 (m, 1H), 3.46 (m, 2H), 3.16 (m, 2H), 2.87 (m, 3H), 2.76 (t, J = 6.83 Hz), 2.70 (m, 2H), 2.49 (m, 4H), 1.87 (m, 1H), 1.66 (m, 4 hr), 1.34 (m, 1 hr), 1.22 (m, 1H), 0.84 (m, 1H), 0.76 (m, 1H). MS (ESI pos) m/z found 662.1 (M + H).

## Conjugation of NAI-AK with Alexa594 azide using CuAAA click chemistry

Compound 4 (0.75 mg, 2.27 mM) in water was added to a solution of AFdye 594 azide (Click Chemistry Tool, Scottsdale, AZ) in methanol (1 mg, 1.6 mM). A freshly-made ascorbate solution (150 µl, 100 mM, adjust pH to 4.0) was added to the mixture and stirred at room temperature. To this solution, a mixture of $CuSO_4$ (12 µl, 25 mM) and BTTES (18 µl, 50 mM) was added and stirring was continued for 2.5 hr. HPLC analysis of the crude material using a gradient solution of 5% to 65% water and acetonitrile containing 0.1% trifluoroacetic acid in 30 min and a reverse phase (Jupiter, 5 µm, C18, 300 Å, 250 × 4.6 mm) showed two major peaks. The peak at 25 min was identified as the product and at 28 min as the free dye by dual wavelength UV-detection (at 220 and 594 nm). The crude material was then purified using a preparative HPLC system using the same gradient condition and a preparative reverse phase column (Jupiter, 5 µm, C18, 300 Å, 250 × 21.2 mm). The purity of product was greater than 95% by HPLC analysis. The collected fractions were combined and lyophilized to obtaind dried powders. The electron-spray ionized mass spectrum confirmed the product peak at 755.2 (m/z +2). The product was dissolved in methanol and measured OD at maximal OD of 591 nm. Using absorption extinction (88,000) of free dye provided by the company (Click Chemistry Tools), the yield was 0.81 mg or 47%. NAI-A488 was prepared in a similar method that yielded 49% product.

## Acknowledgements

The authors thank Drs. Maria Torecilla and Brooks Robinson for their comments and technical help with brain slice preparations. We also thank the Medicinal Chemistry Core (Dr. Aeron Nilson), Flow Cytometry Core (Pamela Canaday) and Bioanalytical Shared Resource/Pharmacokinetic Core (Jenny Luo) at OHSU for their superior services. The work is funded by grants DA008163-JTW, DA048136-JTW, DLF and SA, and DA042779-WTB from the National Institutes of Health. A portion of this work was supported the Intramural Research Programs of the National Institute on Drug Abuse and National Institute of Alcohol Abuse and Alcoholism.

## Additional information

### Funding

| Funder | Grant reference number | Author |
| --- | --- | --- |
| National Institute on Drug Abuse | DA008163 | John T Williams |
| National Institute on Drug Abuse | DA048136 | Seksiri Arttamangkul<br>David L Farrens<br>John T Williams |

| National Institute on Drug Abuse | DA042779 | William T Birdsong |

The funders had no role in study design, data collection and interpretation, or the decision to submit the work for publication.

## Author contributions

Seksiri Arttamangkul, Conceptualization, Resources, Data curation, Formal analysis, Validation, Investigation, Visualization, Methodology, Writing—original draft, Writing—review and editing; Andrew Plazek, Conceptualization, Formal analysis, Validation, Investigation, Methodology, Writing—review and editing; Emily J Platt, Thomas F Murray, William T Birdsong, Formal analysis, Validation, Investigation, Methodology, Writing—review and editing; Haihong Jin, Formal analysis, Validation, Investigation, Methodology; Kenner C Rice, Resources, Writing—review and editing; David L Farrens, Supervision, Writing—review and editing; John T Williams, Formal analysis, Funding acquisition, Validation, Investigation, Methodology, Writing—review and editing

## Author ORCIDs

Seksiri Arttamangkul (iD) https://orcid.org/0000-0002-8815-5124
John T Williams (iD) http://orcid.org/0000-0002-0647-6144

## Ethics

Animal experimentation: All animal uses were conducted in accordance with the National Institutes of Health guidelines and with approval from the Institutional Animal Care and Use Committee (IACUC) protocol #IP00000160 of the Oregon Health & Science University. Rats and mice were anesthetized with isofluorane before euthanized with minimal suffering.

## Decision letter and Author response

Decision letter https://doi.org/10.7554/eLife.49319.027
Author response https://doi.org/10.7554/eLife.49319.028

# Additional files

## Supplementary files

• Transparent reporting form DOI: https://doi.org/10.7554/eLife.49319.025

## Data availability

All data generated or analysed during this study are included in the manuscript and supporting files.

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
