## [Decision Letter]

Thank you for submitting your article "Visualizing endogenous opioid receptors in living neurons using ligand-directed chemistry" for consideration by *eLife*. Your article has been reviewed by three peer reviewers, and the evaluation has been overseen by Peggy Mason as Reviewing Editor and Catherine Dulac as the Senior Editor. The following individuals involved in review of your submission have agreed to reveal their identity: Christopher Evans (Reviewer #2); Meritxell Canals (Reviewer #3).

The reviewers have discussed the reviews with one another and the Reviewing Editor has drafted this decision to help you prepare a revised submission.

All agree that this is an exciting step forward. To be able to visualize mu opioid receptors in situ and still functioning is a tool that is likely to be of great use to many. There was some hesitation however because of two major issues.

First, your ideas on why delta and kappa receptors aren't labeled are plausible but not tested. Value would be added by testing your compound on cell lines transfected with delta and kappa receptors.

Second reviewer #2 raises important points about the efficacy and mechanism of labeling, particularly given the lack of endosome labeling (as also pointed out by reviewer #3) and the CTOP blockade.

With these concerns, we discussed whether to send this article back to you (soft reject) for more experiments with the hope of seeing a re-submission at a later time or to ask for revisions, with the expectation that you would possibly alter analyses, include experiments that are on hand, and discuss the caveats identified by the reviewers. We agreed on the latter course because of the novelty and promise of the method. Of course, if you want to add any of the experiments, please do and we will entertain a re-submission when available.

Reviewer #1:

This study describes a covalent labeling strategy for opioid receptors using ligand-guided proximity chemistry. Although the guide ligand used is not specific for mu opioid receptors, the authors demonstrate empirically that MOR is primarily labeled and that labeled MORs are functional. The authors also demonstrate that the method is suitable for labeling endogenous MOR in brain slices, and they provide a survey of labeling patterns in selected brain regions. The images are striking and, as the authors note, strong labeling of TH-negative processes in midbrain and VTA regions is particularly intriguing.

I think that the reagent described is very interesting and the data reported establish utility of the method and appropriately validate specificity. Considering the importance of understanding MOR connectivity in the brain and the relative paucity of methods suitable for detecting endogenous opioid receptors in tissue, I think this is a major advance and worthy of report in its present form. This is a really nice study.

Reviewer #2:

The paper describes the development of a fluorescent ligand-directed probe, naltrexamine-acylimidazole (NAI), that is able to fluorescently tag opioid receptors (appears only Mu receptors with NAI-A594) with subsequent dissociation of the ligand. This results in convincingly fluorescently labeled Mu receptors that can be visualized. This is a technique that could have broad applicability to GPCRs and complement studies with genetically fluorescently-tagged receptors.

1) NAI should bind to all opioid receptors – and the lack of tagging of delta and kappa has some explanation in the Discussion. However, the reasons given are theoretical and the site of modification on the Mu is currently unknown. Negative tagging data for transfected cell lines with delta and kappa would support the conclusions that delta and kappa cannot be tagged with this reagent. There are potentially other explanations that could modify NAI-A594 opioid receptor labeling in slices.

2) The fact that the hydrophilic Mu antagonist CTOP blocked the tagging is suggestive that only surface receptors are labeled by this strategy – do the authors have data pertinent regarding membrane permeability of the reagent?

3) To play devil's advocate one could envisage that Mu function is in fact impaired by the tagging of the receptor and it is homo-dimer driven endocytosis that is suggesting internalization activity. Certainly mu-receptor dragging to internal compartments has been shown to be triggered by activation of other GPCRs. Since 500nM NAI-A594 only blocks a percentage of agonist activity and 30-100nM is used for labeling, the number of tagged receptors may be well below 50%. This leads to the question of efficacy of the tagging and what proportion of the receptors are being tagged in the slices. Can the authors address the extent of receptor modification and possibility of homo-dimer driven endocytosis of NAI-A594-modified receptors if they were dysfunctional?

4) The NAI is an antagonist that is insensitive to states of the receptor and should label opioid receptors in inactive states (such as surface desensitized) – not clear to me at this point if receptors would be labeled that were targeted for exocytosis or signaling intracellularly. In the Discussion it would have been helpful to have a comparison of benefits and pitfalls of this approach vs. genetically labeling receptors with GFP. Clearly the method has advantages over post-fixing methodologies and maybe more sensitive in comparison to genetic tagging but the pros and cons of the technique should be explicit.

5) The lack of endosome labeling in the slices after agonist internalization and ligand-induced reduction of fluorescence is bothersome given the robust labeling of internalized receptors in the transfected cell lines. Some of the explanations were not convincing. That morphine and naloxone causes the same loss of surface fluorescence is suggestive of competitive binding and questions the mechanism of labeling in the slices as the same as the cell lines. Optimally this could be resolved before publication.

6) In Figure 3B the addition of a THGFP alone would help the identification of TH and NAI labeling and whether or not the punctate staining over the cell body is NIA-generated and not blocked by CTAP.

Reviewer #3:

In the current manuscript Arttamangkul et al., report the generation of a fluorescent probe that allows for the labelling of endogenous opioid receptors. This is a major advancement in our ability to visualise these receptors in their endogenous settings as it does not rely of fluorescent tags on the receptors (which requires genetic manipulation) or fluorescent ligands, which limit the use of the "labelled" receptor. In addition, the chemistry between the opioid antagonist and the fluorophore is based on click chemistry, making the reaction versatile for different fluorophores that may be useful for different imaging approaches. The authors show how this probe labels endogenous receptors in the LC and other areas of the brain and that it allows visualization deeper into the slice, which is a major advantage over the use of antibodies. Importantly, after labelling, the receptors remain functional, a key feature for future studies.

Results:

• First paragraph discusses the synthesis of naltrexamine-acylimidazole – it would be beneficial for the field to describe what a "proper linker" means. How was linker length and structure decided upon?

• Competitive radioligand binding shows that the probe (NAI-A594) has affinity for MOR, DOR and KOR. However, experiments done in slices show a pretty selective labelling of MOR, that is either absent in MOR KO or displaced by CTAP. The authors suggest that this subtype selectivity may be due to the residues exposed for the nucleofilic attack of the fluorophore being only present at MOR. It would be useful to provide a model of the three receptors' structure to illustrate this point.

• The internalization data in neurons is intriguing. While a decrease in A594 fluorescence at the plasma membrane is indicative of a reduction of receptor levels at the cell surface, there seems to be no detectable A594 fluorescence in the cytoplasm (where FlagMOR is detected). Moreover, the decrease in fluorescence is also observed after incubation of ligands that do not internalize the receptor. One would think this could be due to a quenching of the fluorophore due to acidic endosomal pH, however, the authors rule this out as A594 is pH insensitive (though a reference needs to be added here). Thus, a few major questions remain to be addressed to explore the full potential of these probes. What does the decrease in fluorescence at the plasma membrane mean? Why does not A594 label receptors in endosomes. Would labelling with A488 or any other fluorophore help address any of these questions? Can an "internalizing" conformation of the receptor be quenching A594?

---

## [Author Response]

Reviewer #2:[…] 1) NAI should bind to all opioid receptors – and the lack of tagging of delta and kappa has some explanation in the Discussion. However, the reasons given are theoretical and the site of modification on the Mu is currently unknown. Negative tagging data for transfected cell lines with delta and kappa would support the conclusions that delta and kappa cannot be tagged with this reagent. There are potentially other explanations that could modify NAI-A594 opioid receptor labeling in slices.

Yes, NAI binds to mu, delta and kappa receptors as shown in competitive radioligand binding assays (Table 1). Experiments using HEK293 expressed FlagMOR, FlagDOR and FlagKOR showed that the labeling of NAI-A594 to FlagDOR was comparable to FlagMOR but was minimal to FlagKOR. We included the results in the manuscript and as the supplementary figure (Figure 2—figure supplement 1). The selectivity toward mu opioid receptors was more apparent with endogenous receptors in brain slices. The results were confirmed with the use of CTAP and were included in the supplementary figure (Figure 4—figure supplement 1).

2) The fact that the hydrophilic Mu antagonist CTOP blocked the tagging is suggestive that only surface receptors are labeled by this strategy – do the authors have data pertinent regarding membrane permeability of the reagent?

NAI-A594 only labels surface receptors. It is reported that opiate alkaloids including naltrexamine are able to penetrate plasma membrane and interact with cytoplasmic receptors (Stoeber et al., 2018). Alexa 594 is hydrophilic and negatively charged due to sulfonation thus once tagged to naltrexamine will decrease plasma membrane permeability. Moreover, acylimidazole reactive group is shown to label only plasma membrane proteins (Tamura and Hamachi, 2019). Generally, low concentrations of NAI-A594 (30 to 100 nM) were used to sufficiently label the receptors. If NAI-A594 can penetrate the plasma membrane, it would take a much longer incubation time to label the receptors. A much higher concentration (10 μM used in Stoeber et al., 2018) would be necessary. The drawback for using fluorescent probes at this high concentration will be the significant non-specific background. Some fluorescent puncta in the cytoplasm of HEK293 may occur due to the high expression of receptors resulting in constitutive internalization. Additionally, we did not observe any cytoplasmic labeling of neurons in brain slices even after more than 2 hours of incubation (see Figure 3—figure supplement 1). The fluorescent puncta found in LC neurons from THGFP mice was autofluorescence that also appeared in CTAP treated cells. We included the separated GFP and NAI-A594 channels of Figure 3B(d) in the supplement (Figure 3—figure supplement 2) to show these autofluorescence.

3) To play devil's advocate one could envisage that Mu function is in fact impaired by the tagging of the receptor and it is homo-dimer driven endocytosis that is suggesting internalization activity. Certainly mu-receptor dragging to internal compartments has been shown to be triggered by activation of other GPCRs. Since 500nM NAI-A594 only blocks a percentage of agonist activity and 30-100nM is used for labeling, the number of tagged receptors may be well below 50%. This leads to the question of efficacy of the tagging and what proportion of the receptors are being tagged in the slices. Can the authors address the extent of receptor modification and possibility of homo-dimer driven endocytosis of NAI-A594-modified receptors if they were dysfunctional?

The concentration and time dependence of labeling has been done. Longer incubation times increase binding (fluorescence). The increased fluorescence could be interpreted in two ways. One is more receptors have been labeled. It is also possible that more sites on the receptor are tagged. During the incubation the binding pocket becomes available for a new molecule of NAI-A594. The length and flexibility of the linker may result in the labeling of new sites. Fluorescence intensity does not increase after incubating brain slices longer than 2 hours. (left slice in the vial for 3-4 hours give similar staining to 2 hours).

LC slices have been incubated with NAI-A594 (300 nM and 1 µM) for 2 hours and still did not observe any receptor internalization.

Increasing the concentration of NAI-A594 to 500 nM (5 min) blocked only a small percent of opioid activity but there was strong labeling following this treatment (see Figure 7). The minimal blockage of NAI-A594 suggested that the compound itself has a low affinity (also suggested in the binding assays) and was competitively displaced by ME.

4) The NAI is an antagonist that is insensitive to states of the receptor and should label opioid receptors in inactive states (such as surface desensitized) – not clear to me at this point if receptors would be labeled that were targeted for exocytosis or signaling intracellularly. In the Discussion it would have been helpful to have a comparison of benefits and pitfalls of this approach vs. genetically labeling receptors with GFP. Clearly the method has advantages over post-fixing methodologies and maybe more sensitive in comparison to genetic tagging but the pros and cons of the technique should be explicit.

Pros: NAI-A594 labels plasma membrane receptors and excludes receptors during exocytosis or signaling intracellularly. This has an advantage in identifying the mature receptors with correct folding that are capable of binding to opioid ligands. The GFP tagged receptors report a mixture of functional and non-functional unfolded receptors. We have included this comparison in the Discussion.

Cons: NAI-A594 cannot be used to label cytoplasmic receptors but other fluorophores that are more lipophilic such as BODIPY dyes which are smaller and non-charged molecules compared to Alexa dyes may be used to help permeabilize into cells.

5) The lack of endosome labeling in the slices after agonist internalization and ligand-induced reduction of fluorescence is bothersome given the robust labeling of internalized receptors in the transfected cell lines. Some of the explanations were not convincing. That morphine and naloxone causes the same loss of surface fluorescence is suggestive of competitive binding and questions the mechanism of labeling in the slices as the same as the cell lines. Optimally this could be resolved before publication.

A portion of labeling is competitive. This was shown in HEK293 cells as well. Flow cytometry results showed that a portion of labeling was displaced by naloxone (Figure 2B). The competitive labeling occurs when NAI-A594 occupies the binding site even on receptors that have been previously covalently linked with the fluorescent ligand.

6) In Figure 3B the addition of a THGFP alone would help the identification of TH and NAI labeling and whether or not the punctate staining over the cell body is NIA-generated and not blocked by CTAP.

Images are now shown separately in a new supplementary figure (Figure 3—figure supplement 2). An explanation of issues associated with autofluorescence is also included in the Results.

Reviewer #3:[…] Results:• First paragraph discusses the synthesis of naltrexamine-acylimidazole – it would be beneficial for the field to describe what a "proper linker" means. How was linker length and structure decided upon?

The design strategy and possible lengths of the linker are now discussed. The design was based upon the crystal structure of β-funaltrexamine (β-FNA) bound MOR. The key covalent attachment of β-FNA at K233 side-chain was purposely avoided. The linker was designed to be longer for NAI compared to β-FNA.

• Competitive radioligand binding shows that the probe (NAI-A594) has affinity for MOR, DOR and KOR. However, experiments done in slices show a pretty selective labelling of MOR, that is either absent in MOR KO or displaced by CTAP. The authors suggest that this subtype selectivity may be due to the residues exposed for the nucleofilic attack of the fluorophore being only present at MOR. It would be useful to provide a model of the three receptors' structure to illustrate this point.

The speculation is that the longer linker will find nucleophilic side chains of amino acids on extracellular loops, both loops 2 and 3. There are several sites that can be modified, including Ser, Tyr, Thr and Lys. The different of these sites on MOR, DOR and KOR were summarized and presented in the supplementary figure (Figure 1—figure supplement 1).

• The internalization data in neurons is intriguing. While a decrease in A594 fluorescence at the plasma membrane is indicative of a reduction of receptor levels at the cell surface, there seems to be no detectable A594 fluorescence in the cytoplasm (where FlagMOR is detected). Moreover, the decrease in fluorescence is also observed after incubation of ligands that do not internalize the receptor. One would think this could be due to a quenching of the fluorophore due to acidic endosomal pH, however, the authors rule this out as A594 is pH insensitive (though a reference needs to be added here).

The reference has been added.

Thus, a few major questions remain to be addressed to explore the full potential of these probes. What does the decrease in fluorescence at the plasma membrane mean?

1) The decrease is caused by competitive binding component.

2) Quenching as a result of conformation change that allow energy transfer between Alexa dye and Trp nearby. We don’t think this is the case because the decrease fluorescence was not reversible after washing the ligand.

Why does not A594 label receptors in endosomes.

A594 tagged receptors were clearly found in endosomes after agonist activation in HEK293 cells and primary culture of habenula neurons (Damien Julie, UCSF, personal communication). In brain slices, problems of autofluorescence and the thickness of slices reduce sensitivity of the fluorescence signal such that identification of labeled endosomes has not been possible. It is also possible that MORs in brain slices may recycle faster and rapidly redistribute laterally. Observing endosomes using labeled antibodies is possible, perhaps because each antibody has multiple fluorescent dyes. It could also be possible that the recycling of the antibody labeled receptor is slowed allowing the observation of endosomes.

Would labelling with A488 or any other fluorophore help address any of these questions? Can an "internalizing" conformation of the receptor be quenching A594?

Energy transfer quenching is possible after internalization. Changing to a fluorophore that is not sensitive to this phenomenon would test this hypothesis.

References:

Stoeber et al. (2018) A genetically encoded biosensor reveals location bias of opioid drug action. Neuron 98:963-976.